# *Multi-Prize Lottery Ticket Hypothesis*:
# FINDING ACCURATE BINARY NEURAL NETWORKS BY PRUNING A RANDOMLY WEIGHTED NETWORK

**James Diffenderfer & Bhavya Kailkhura**
Center for Applied Scientific Computing
Lawrence Livermore National Laboratory
Livermore, CA 94550, USA
{diffenderfer2,kailkhura1}@llnl.gov

## ABSTRACT

Recently, Frankle & Carbin (2019) demonstrated that randomly-initialized dense networks contain subnetworks that once found can be trained to reach test accuracy comparable to the trained dense network. However, finding these high performing trainable subnetworks is expensive, requiring iterative process of training and pruning weights. In this paper, we propose (and prove) a stronger *Multi-Prize Lottery Ticket Hypothesis*:

*A sufficiently over-parameterized neural network with random weights contains several subnetworks (winning tickets) that (a) have comparable accuracy to a dense target network with learned weights (prize 1), (b) do not require any further training to achieve prize 1 (prize 2), and (c) is robust to extreme forms of quantization (i.e., binary weights and/or activation) (prize 3).*

This provides a new paradigm for learning compact yet highly accurate binary neural networks simply by pruning and quantizing randomly weighted full precision neural networks. We also propose an algorithm for finding multi-prize tickets (MPTs) and test it by performing a series of experiments on CIFAR-10 and ImageNet datasets. Empirical results indicate that as models grow deeper and wider, multi-prize tickets start to reach similar (and sometimes even higher) test accuracy compared to their significantly larger and full-precision counterparts that have been weight-trained. Without ever updating the weight values, our MPTs-1/32 not only set new binary weight network state-of-the-art (SOTA) Top-1 accuracy – 94.8% on CIFAR-10 and 74.03% on ImageNet – but also outperform their full-precision counterparts by 1.78% and 0.76%, respectively. Further, our MPT-1/1 achieves SOTA Top-1 accuracy (91.9%) for binary neural networks on CIFAR-10. Code and pre-trained models are available at: `https://github.com/chrundle/biprop`.

## 1 INTRODUCTION

Deep learning (DL) has made a significant breakthroughs in a wide range of applications (Goodfellow et al., 2016). These performance improvements can be attributed to the significant growth in the model size and the availability of massive computational resources to train such models. Therefore, these gains have come at the cost of large memory consumption, high inference time, and increased power consumption. This not only limits the potential applications where DL can make an impact but also have some serious consequences, such as, (a) generating huge carbon footprint, and (b) creating roadblocks to the democratization of AI. Note that significant parameter redundancy and a large number of floating-point operations are key factors incurring the these costs. Thus, for discarding the redundancy from DNNs, one can either (a) *Prune:* remove non-essential connections from an existing dense network, or (b) *Quantize:* constrain the full-precision (FP) weight and activation values to a set of discrete values which allows them to be represented using fewer bits. Further, one can exploit the complementary nature of pruning and quantization to combine their strengths.

| | Configuration | Model | Weights | Memory Savings | Comp. Savings | Params | Accuracy |
|---|---|---|---|---|---|---|---|
| CIFAR-10 | Dense (32/32) | ResNet18 | Learned | 1× | 1× | 11.2 M | 93.02% |
| | ProxQuant (1/32) | ResNet56 | Learned | ~32× | ~2× | 0.85 M | 92.3% |
| | MPT (1/32) | ResNet18 | Pruned | ~32× | ~2× | 2.2 M | 94.8% |
| ImageNet | Dense (32/32) | ResNet34 | Learned | 1× | 1× | 21.8 M | 73.27% |
| | Quant-Net (1/32) | ResNet50 | Learned | ~32× | ~2× | 25.6 M | 72.8% |
| | MPT (1/32) | WRN-50 | Pruned | ~32× | ~2× | 13.7 M | 74.03% |

Figure 1: **Multi-Prize Ticket Performance**: Multi-prize tickets, obtained only by pruning and binarizing random networks, outperforms trained full precision and SOTA binary weight networks.

Although pruning and quantization[1] are typical approaches used for compressing DNNs (Neill, 2020), it is not clear under what conditions and to what extent compression can be achieved without sacrificing the accuracy. The most extreme form of quanitization is binarization, where weights and/or activations can only have two possible values, namely $-1(0)$ or $+1$ (the interest of this paper). In addition to saving memory, binarization results in more power efficient networks with significant computation acceleration since expensive multiply-accumulate operations (MACs) can be replaced by cheap XNOR and bit-counting operations (Qin et al., 2020a). In light of these benefits, it is of interest to question if conditions exists such that a binarized DNN can be pruned to achieve accuracy comparable to the dense FP DNN. More importantly, even if these favourable conditions are met then how do we find these extremely compressed (or compact) and highly accurate subnetworks?

Traditional pruning schemes have shown that a pretrained DNN can be pruned without a significant loss in the performance. Recently, (Frankle & Carbin, 2019) made a breakthrough by showing that dense network contain sparse subnetworks that can match the performance of the original network when trained from scratch with weights being reset to their initialization (*Lottery Ticket Hypothesis*). Although the original approach to find these subnetworks still required training the dense network, some efforts (Wang et al., 2020b; You et al., 2019; Wang et al., 2020a) have been carried out to overcome this limitation. Recently a more intriguing phenomenon has been reported – a dense network with random initialization contains subnetworks that achieve high accuracy, without any further training (Zhou et al., 2019; Ramanujan et al., 2020; Malach et al., 2020; Orseau et al., 2020). These trends highlight good progress being made towards *efficiently* and *accurately* pruning DNNs.

In contrast to these positive developments for pruning, results on binarizing DNNs have been mostly negative. To the best of our knowledge, post-training schemes have not been successful in binarizing pretrained models without retraining. Even with training binary neural networks (BNNs) from scratch (though inefficient), the community has not been able to make BNNs achieve comparable results to their full precision counterparts. The main reason being that network structures and weight optimization techniques are predominantly developed for full precision DNNs and may not be suitable for training BNNs. Thus, closing the gap in accuracy between the full precision and the binarized version may require a paradigm shift. Furthermore, this also makes one wonder if *efficiently* and *accurately* binarizing DNNs similar to the recent trends in pruning is ever feasible.

In this paper, we show that a randomly initialized dense network contains extremely sparse binary subnetworks that without any weight training (i.e., *efficient*) have comparable performance to their trained dense and full-precision counterparts (i.e., *accurate*). Based on this, we state our hypothesis:

> **Multi-Prize Lottery Ticket Hypothesis.** *A sufficiently over-parameterized neural network with random weights contains several subnetworks (winning tickets) that (a) have comparable accuracy to a dense target network with learned weights (prize 1), (b) do not require any further training to achieve prize 1 (prize 2), and (c) is robust to extreme forms of quantization (i.e., binary weights and/or activation) (prize 3).*

**Contributions.** First, we propose the multi-prize lottery ticket hypothesis as a new perspective on finding neural networks with drastically reduced memory size, much faster test-time inference and

---

[1]A detailed discussion on related work on pruning and quantization is provided in Appendix F.

lower power consumption compared to their dense and full-precision counterparts. Next, we provide theoretical evidence of the existence of highly accurate binary subnetworks within a randomly weighted DNN (i.e., proving the multi-prize lottery ticket hypothesis). Specifically, we mathematically prove that we can find an $\varepsilon$-approximation of a fully-connected ReLU DNN with width $n$ and depth $\ell$ using a sparse binary-weight DNN of sufficient width. Our proof indicates that this can be accomplished by pruning and binarizing the weights of a randomly weighted neural network that is a factor $O(n^{3/2}\ell/\varepsilon)$ wider and $2\ell$ deeper. To the best of our knowledge, this is the first theoretical work proving the existence of highly accurate binary subnetworks within a sufficiently overparameterized randomly initialized neural network. Finally, we provide **biprop** (**bi**narize-**pr**une **op**timizer) in Algorithm 1 to identify MPTs within randomly weighted DNNs and empirically test our hypothesis. This provides a completely new way to learn BNNs without relying on weight-optimization.

**Results.** We explore two variants of multi-prize tickets – one with binary weights (MPT-1/32) and other with binary weights and activation (MPT-1/1) where $x/y$ denotes $x$ and $y$ bits to represent weights and activation, respectively. MPTs we find have $60 - 80\%$ fewer parameters than the original network. We perform a series of experiments on on small and large scale datasets for image recognition, namely CIFAR-10 (Krizhevsky et al., 2009) and ImageNet (Deng et al., 2009). On CIFAR-10, we test the performance of multi-prize tickets against the trend of making the model deeper and wider. We found that as models grow deeper and wider, both variants of multi-prize tickets start to reach similar (and sometimes even higher) test accuracy compared to the dense and full precision original network with learned weights. In other words, the performance of multi-prize tickets improves with the amount of redundancy in the original network. We also carry out experiments with state-of-the-art (SOTA) architectures on CIFAR-10 and ImageNet datasets with an aim to investigate their redundancy. We find that within most randomly weighted SOTA DNNs reside extremely compact (i.e., sparse and binary) subnetworks which are smaller than, but match the performance of trained target dense and full precision networks. Furthermore, with minimal hyperparameter tuning, our MPTs achieve Top-1 accuracy comparable to (or higher than) SOTA BNNs. The performance of MPTs is further improved by allowing the parameters in BatchNorm layer to be learned. Finally, on both CIFAR-10 and ImageNet, MPT-1/32 subnetworks outperform their significantly larger and full-precision counterparts that have been weight-trained.

## 2 MULTI-PRIZE LOTTERY TICKETS: THEORY AND ALGORITHMS

We first prove the existence of MPTs in an overparameterized randomly weighted DNN. For ease of presentation, we state an informal version of Theorem 2 which can be found in Appendix B. We then explore two variants of tickets (MPT-1/32 and MPT-1/1) and provide an algorithm to find them.

### 2.1 PROVING THE MULTI-PRIZE LOTTERY TICKETS HYPOTHESIS

In this section we seek to answer the following question: *What is the required amount of overparameterization such that a randomly weighted neural network can be compressed to a sparse binary subnetwork that approximates a dense trained target network?*

**Theorem 1.** *(Informal Statement of Theorem 2) Let $\varepsilon, \delta > 0$. For every fully-connected (FC) target network with ReLU activations of depth $\ell$ and width $n$ with bounded weights, a random binary FC network with ReLU activations of depth $2\ell$ and width $O\left((\ell n^{3/2}/\varepsilon) + \ell n \log(\ell n/\delta)\right)$ contains with probability $(1 - \delta)$ a binary subnetwork that approximates the target network with error at most $\varepsilon$.*

*Sketch of Proof.* Consider a FC ReLU network $F(\boldsymbol{x}) = \boldsymbol{W}^{(\ell)}\sigma(\boldsymbol{W}^{(\ell-1)}\cdots\sigma(\boldsymbol{W}^{(1)}\boldsymbol{x}))$, where $\sigma(x) = \max\{0, x\}$, $\boldsymbol{x} \in \mathbb{R}^d$, $\boldsymbol{W}^{(i)} \in \mathbb{R}^{k_i \times k_{i-1}}$, $k_0 = d$, and $i \in [\ell]$. Additionally, consider a FC network with binary weights given by $G(\boldsymbol{x}) = \boldsymbol{B}^{(\ell')}\sigma(\boldsymbol{B}^{(\ell'-1)}\cdots\sigma(\boldsymbol{B}^{(1)}\boldsymbol{x}))$, where $\boldsymbol{B}^{(i)} \in \{-1, +1\}^{k_i' \times k_{i-1}'}$, $k_0' = d$, and $i \in [\ell']$. Our goal is to determine a lower bound on the depth, $\ell'$, and the widths, $\{k_i'\}_{i=1}^{\ell'}$, such that with probability $(1 - \delta)$ the network $G(\boldsymbol{x})$ contains a subnetwork $\tilde{G}(\boldsymbol{x})$ satisfying $\|\tilde{G}(\boldsymbol{x}) - F(\boldsymbol{x})\| \leq \varepsilon$, for any $\varepsilon > 0$ and $\delta \in (0, 1)$. We first establish lower bounds on the width of a network of the form $\boldsymbol{g}(\boldsymbol{x}) = \boldsymbol{B}^{(2)}\sigma(\boldsymbol{B}^{(1)}\boldsymbol{x})$ such that with probability $(1 - \delta')$ there exists a subnetwork $\tilde{\boldsymbol{g}}(\boldsymbol{x})$ of $\boldsymbol{g}(\boldsymbol{x})$ s.t. $\|\tilde{\boldsymbol{g}}(\boldsymbol{x}) - \sigma(\boldsymbol{W}\boldsymbol{x})\| \leq \varepsilon'$, for any $\varepsilon' > 0$ and $\delta' \in (0, 1)$. This process is carried out in detail in Lemmas 1, 2, and 3 in Appendix B. We have

now approximated a single layer FC real-valued network using a subnetwork of a two-layer FC binary network. Hence, we can take $\ell' = 2\ell$ and Lemma 3 provides lower bounds on the width of each intermediate layer such that with probability $(1 - \delta)$ there exists a subnetwork $\tilde{G}(\boldsymbol{x})$ of $G(\boldsymbol{x})$ satisfying $\|\tilde{G}(\boldsymbol{x}) - F(\boldsymbol{x})\| \leq \varepsilon$. This is accomplished in Theorem 2 in Appendix B. $\qquad\square$

To the best of our knowledge this is the first theoretical result proving that a sparse binary-weight DNN that can approximate a real-valued target DNN. As it has been established that real-valued DNNs are universal approximators (Scarselli & Tsoi, 1998), our result carries the implication that sparse binary-weight DNNs are also universal approximators. In relation to the first result establishing the existence of real-valued subnetworks in a randomly weighted DNN approximating a real-valued target DNN (Malach et al., 2020), the lower bound on the width established in Theorem 2 is better than their lower bound of $O\left(\ell^2 n^2 \log(\ell n/\delta)/\varepsilon^2\right)$.

## 2.2 FINDING MULTI-PRIZE WINNING TICKETS

Given the existence of multi-prize winning tickets from Theorem 2, a natural question arises – *How should we find them?* In this section, we answer this question by introducing an algorithm for finding multi-prize tickets.[2] Specifically, we explore two variants of multi-prize tickets in this paper – 1) MPT-1/32 where weights are quantized to 1-bit with activations being real valued (i.e., 32-bits) and 2) MPT-1/1 where both weights and activations are quantized to 1-bit. We first outline a generic process for identifying MPTs along with some theoretical motivation for our approach.

Given a neural network $g(\boldsymbol{x}; \boldsymbol{W})$ with weights $\boldsymbol{W} \in \mathbb{R}^m$, we can express a subnetwork of $g$ using a binary mask $\boldsymbol{M} \in \{0, 1\}^m$ as $g(\boldsymbol{x}; \boldsymbol{M} \odot \boldsymbol{W})$, where $\odot$ denotes the Hadamard product. Hence, a binary subnetwork can be expressed as $g(\boldsymbol{x}; \boldsymbol{M} \odot \boldsymbol{B})$, where $\boldsymbol{B} \in \{-1, +1\}^m$. Lemma 1 in Appendix B indicates that rescaling the binary weights to $\{-\alpha, \alpha\}$ using a gain term $\alpha \in \mathbb{R}$ is necessary to achieve good performance of the resulting subnetwork. We note that the use of gain terms is common in binary neural networks (Qin et al., 2020a; Martinez et al., 2020; Bulat & Tzimiropoulos, 2019). Combining all this allows us to represent a binary subnetwork as $g(\boldsymbol{x}; \alpha(\boldsymbol{M} \odot \boldsymbol{B}))$.

Now we focus on how to update $\boldsymbol{M}$, $\boldsymbol{B}$, and $\alpha$. Suppose $f(\boldsymbol{x}; \boldsymbol{W}^*)$ is a target network with optimized weights $\boldsymbol{W}^*$ that we wish to approximate. Assuming $g(\boldsymbol{x}; \cdot)$ is $\kappa$-Lipschitz continuous yields

$$\underbrace{\|g\left(\boldsymbol{x}; \alpha(\boldsymbol{M} \odot \boldsymbol{B})\right) - f(\boldsymbol{x}; \boldsymbol{W}^*)\|}_{\text{MPT error}} \leq \kappa \underbrace{\|\boldsymbol{M} \odot (\boldsymbol{W} - \alpha\boldsymbol{B})\|}_{\text{Binarization error}} + \underbrace{\|g(\boldsymbol{x}; \boldsymbol{M} \odot \boldsymbol{W}) - f(\boldsymbol{x}; \boldsymbol{W}^*)\|}_{\text{Subnetwork error}}. \quad (1)$$

Hence, the MPT error is bounded above by the error of the subnetwork of $g$ with the original weights and the error from binarizing the current subnetwork. This informs our approach for identifying MPTs: 1) Update a pruning mask $\boldsymbol{M}$ that reduces the subnetwork error (lines $7 - 9$ in Algorithm 1), and 2) apply binarization with a gain term that minimizes the binarization error (lines 4 and 10).

We first discuss how to update $\boldsymbol{M}$. While we could search for $\boldsymbol{M}$ by minimizing the subnetwork error in (1), this would require the use of a pretrained target network (i.e., $f(\boldsymbol{x}; \boldsymbol{W}^*)$). To avoid requiring a target network in our method we instead aim to minimize the training loss w.r.t. $\boldsymbol{M}$ in the current binary subnetwork. Directly optimizing over the pruning mask is a combinatorial problem. So to update the pruning mask efficiently we optimize over a set of scores $\boldsymbol{S} \in \mathbb{R}^m$ corresponding to each randomly initialized weight in the network. In this approach, each component of the randomly initialized weights is assigned a pruning score. The pruning scores are updated via backpropagation by computing the gradient of the loss function over minibatches with respect to the pruning scores (line 7). Then the magnitude of the scores in absolute value are used to identify the $P$ percent of weights in each layer that are least important to the success of the binary subnetwork (line 8). The components of the pruning mask corresponding to these indices are set to 0 and the remaining components are set to 1 (line 9). To avoid unintentionally pruning an entire layer of the network, we use a pruning mask for each layer that prunes $P$ percent of the weights in that layer. The choice to use pruning scores to update the mask $\boldsymbol{M}$ was due to the fact that it is computationally efficient. The use of pruning scores is a well-established optimization technique used in a range of applications (Joshi & Boyd, 2009; Ramanujan et al., 2020).

---

[2]Although our results are derived under certain assumptions (e.g., fully-connected, ReLU neural network approximated by a subnetwork with binary weights), our algorithm is not restricted by these assumptions.

---

**Algorithm 1 biprop**: Finding multi-prize tickets in a randomly weighted neural network

---

1: **Input**: Neural network $g(\boldsymbol{x}; \cdot)$ with 1- or 32-bit activations; Network depth $\ell$; Layer widths $\{k_j\}_{j=1}^{\ell}$; Loss function $L$; Training data $\{(\boldsymbol{x}^{(i)}, \boldsymbol{y}^{(i)})\}_{i=1}^{N}$; Pruning percentage $P$.

2: *Randomly Initialize FP Parameters*: Network weights $\{\boldsymbol{W}^{(j)}\}_{j=1}^{\ell}$; Pruning scores $\{\boldsymbol{S}^{(j)}\}_{j=1}^{\ell}$.

3: *Initialize Layerwise Pruning Masks*: $\{\boldsymbol{M}^{(j)}\}_{j=1}^{\ell}$ each to $\mathbf{1}$.

4: *Initialize Binary Subnetwork Weights*: $\{\boldsymbol{B}^{(j)}\}_{j=1}^{\ell} \leftarrow \{\mathrm{sign}(\boldsymbol{W}^{(j)})\}_{j=1}^{\ell}$.

5: *Initialize Layerwise Gain Terms*: $\{\alpha^{(j)}\}_{j=1}^{\ell} \leftarrow \{\|\boldsymbol{M}^{(j)} \odot \boldsymbol{W}^{(j)}\|_1 / \|\boldsymbol{M}^{(j)}\|_1\}_{j=1}^{\ell}$.

6: **for** $k = 1$ to $N_{epochs}$ **do**

7: $\quad \boldsymbol{S}^{(j)} \leftarrow \boldsymbol{S}^{(j)} - \eta \nabla_{\boldsymbol{S}^{(j)}} L(\{\alpha^{(j)}(\boldsymbol{M}^{(j)} \odot \boldsymbol{B}^{(j)})\}_{j=1}^{\ell})$ $\qquad$ Update pruning scores at layer $j$

8: $\quad \{\tau(i)\}_{i=1}^{k_j} \leftarrow$ Sorting of indices $\{i\}_{i=1}^{k_j}$ s.t. $|\boldsymbol{S}_{\tau(i)}^{(j)}| \leq |\boldsymbol{S}_{\tau(i+1)}^{(j)}|$ Index sort over values $|\boldsymbol{S}^{(j)}|$

9: $\quad \boldsymbol{M}_i^{(j)} \leftarrow \mathbb{1}_{\{\tau(i) \geq \lceil k_j P / 100 \rceil\}}(i)$ $\qquad$ Update pruning mask at layer $j$

10: $\quad \alpha^{(j)} \leftarrow \|\boldsymbol{M}^{(j)} \odot \boldsymbol{W}^{(j)}\|_1 / \|\boldsymbol{M}^{(j)}\|_1$ $\qquad$ Update gain term at layer $j$

11: **Output**: Return Binarized Subnetwork $g(\boldsymbol{x}; \{\alpha^{(j)}(\boldsymbol{M}^{(j)} \odot \boldsymbol{B}^{(j)})\}_{j=1}^{\ell})$.

---

We now consider how to update $\boldsymbol{B}$ and $\alpha$. By keeping $\boldsymbol{M}$ fixed, we can derive the following closed form expressions that minimize the binarization error in (1): $\boldsymbol{B}^* = \mathrm{sign}(\boldsymbol{W})$ and $\alpha^* = \|\boldsymbol{M} \odot \boldsymbol{W}\|_1 / \|\boldsymbol{M}\|_1$. These closed form expressions indicate that only the gain term needs to be recomputed after each update to $\boldsymbol{M}$. Hence, $\boldsymbol{B} = \mathrm{sign}(\boldsymbol{W})$ throughout our entire approach (line 4). We update a gain term for each layer of the subnetwork in our approach based on the formula for $\alpha^*$ (line 10). More details on the derivation of $\boldsymbol{B}^*$ and $\alpha^*$ are provided in Appendix C.

Pseudocode for our method **biprop** (**bi**narize-**pr**une **op**timizer) is provided in Algorithm 1 and cross-entropy loss is used in our experiments. Note that the process for identifying MPT-1/32 and MPT-1/1 differs only in computation of the gradient. Next, we explain how these gradients can be computed.

### 2.2.1 Updating Pruning Scores for Binary-Weight Tickets (MPT-1/32)

As an example, for a FC network where the state at each layer is defined recursively by $\boldsymbol{U}^{(1)} = \alpha^{(1)}(\boldsymbol{B}^{(1)} \odot \boldsymbol{M}^{(1)})\boldsymbol{x}$ and $\boldsymbol{U}^{(j)} = \alpha^{(j)}(\boldsymbol{B}^{(j)} \odot \boldsymbol{M}^{(j)})\sigma(\boldsymbol{U}^{(j-1)})$ we have $\frac{\partial L}{\partial S_{p,q}^{(j)}} = \frac{\partial L}{\partial U_q^{(j)}} \frac{\partial U_q^{(j)}}{\partial M_{p,q}^{(j)}} \frac{\partial M_{p,q}^{(j)}}{\partial S_{p,q}^{(j)}}$. We use the straight-through estimator (Bengio et al., 2013) for $\frac{\partial M_{p,q}^{(j)}}{\partial S_{p,q}^{(j)}}$ which yields $\frac{\partial L}{\partial S_{p,q}^{(j)}} = \frac{\partial L}{\partial U_q^{(j)}} \alpha^{(j)} B_{p,q}^{(j)} \sigma\left(U_p^{(j-1)}\right)$, where $\frac{\partial L}{\partial U_q^{(j)}}$ is computed via backpropagation.

### 2.2.2 Updating Pruning Scores for Binary-Activation Tickets (MPT-1/1)

Note that MPT-1/1 uses the sign activation function. From Section 2.2.1, it immediately follows that $\frac{\partial L}{\partial S_{p,q}^{(j)}} = \frac{\partial L}{\partial U_q^{(j)}} \alpha^{(j)} B_{p,q}^{(j)} \mathrm{sign}\left(U_p^{(j-1)}\right)$. However, updating $\frac{\partial L}{\partial U_q^{(j)}}$ via backpropagation requires a gradient estimator for the sign activation function. To motivate our choice of estimator note that we can approximate the sign function using a quadratic spline parameterized by some $t > 0$:

$$s_t(x) = \begin{cases} -1 & : & x < -t \\ q_1(x) & : & x \in [-t, 0) \\ q_2(x) & : & x \in [0, t) \\ 1 & : & x \geq t \end{cases} . \qquad (2)$$

In (2), $q_i(x) = a_i x^2 + b_i x + c_i$ and suitable values for the coefficients are derived using the following zero- and first-order constraints: $q_1(-t) = -1$, $q_1(0) = 0$, $q_2(0) = 0$, $q_2(t) = 1$, $q_1'(-t) = 0$, $q_1'(0) = q_2'(0)$, and $q_2'(t) = 0$. This yields $q_1(x) = (x/t)^2 + 2(x/t)$ and $q_2(x) = -(x/t)^2 + 2(x/t)$. As $s_t(x)$ approximates $\mathrm{sign}(x)$, we can use $s_t'(x)$ as our gradient estimator. Since $q_1'(x) = \frac{2}{t}(1 + \frac{x}{t})$ and $q_2'(x) = \frac{2}{t}(1 - \frac{x}{t})$ it follows that $s_t'(x) = \left[\frac{2}{t}\left(1 - \frac{|x|}{t}\right)\right] \mathbb{1}_{\{x \in [-t,t]\}}(x)$. The choice to approximate sign using a quadratic spline instead of a cubic spline results in a gradient estimator that can be implemented efficiently in PyTorch

as `torch.clamp(2*(1-torch.abs(x)/t)/t,min=0.0)`. We note that $\lim_{t \to 0} s_t(x) = \text{sign}(x)$, which suggests that smaller values of $t$ yield more suitable approximations. Our experiments use $s'_1(x)$ as the gradient estimator since we found it to work well in practice. Finally, we note that taking $t = 1$ in our gradient estimator yields the same value as the gradient estimator in (Liu et al., 2018a), however, our implementation in PyTorch is $6\times$ more memory efficient.

## 3 EXPERIMENTAL RESULTS

The primary goal of the experiments in Section 3.1 is to empirically verify our *Multi-Prize Lottery Ticket Hypothesis*. As a secondary objective, we would like to determine tunable factors that make randomly-initialized networks amenable to containing readily identifiable Multi-Prize Tickets (MPTs). Thus, we test our hypothesis against the general trend of increasing the model size (depth and width) and monitor the accuracy of the identified MPTs. After verifying our *Multi-Prize Lottery Ticket Hypothesis*, we consider the performance of MPTs compared to state-of-the-arts in binary neural networks and their dense counterparts on CIFAR-10 and ImageNet datasets in Section 3.2. Building upon edge-popup (Ramanujan et al., 2020), we implement Algorithm 1 to identify MPTs.[3]

### 3.1 WHERE CAN WE EXPECT TO FIND MULTI-PRIZE TICKETS?

In this section, we empirically test the effect of overparameterization on the performance of MPTs. We overparameterize networks by making them (a) deeper (Sec. 3.1.1) and (b) wider (Sec. 3.1.2).

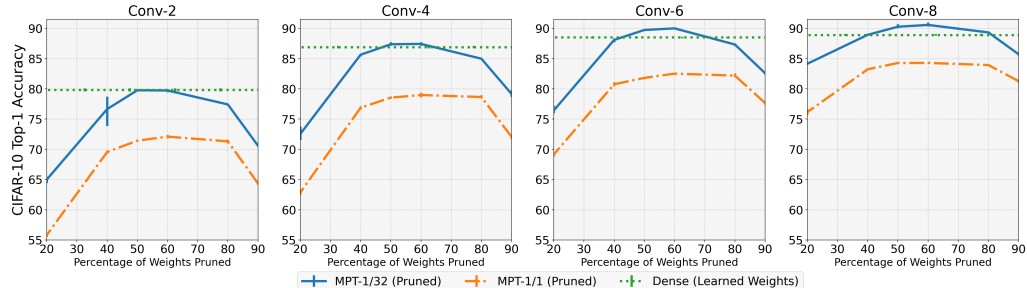

Figure 2: **Effect of Varying Depth and Pruning Rate**: Comparing the Top-1 accuracy of small and binary MPTs to a large, full-precision, and weight-optimized network on CIFAR-10.

We use VGG (Simonyan & Zisserman, 2014) variants as our network architectures for searching for MPTs. In each randomly weighted network, we find winning tickets MPT-1/32 and MPT-1/1 for different pruning rates using Algorithm 1. We choose our baselines as dense full-precision models with learned weights. In all experiments, we use three independent initializations and report the average of Top-1 accuracy with with error bars extending to the lowest and highest Top-1 accuracy. Additional experiment configuration details are provided in Appendix A.

### 3.1.1 DO WINNING TICKETS EXIST IN DEEP NETWORKS?

In this experiment, we empirically test the following hypothesis: *As a network grows deeper, the performance of multi-prize tickets in the randomly initialized network will approach the performance of the same network with learned weights. We are further interested in exploring the required network depth for our hypothesis to be true.*

In Figure 2, we vary the depth of VGG architectures ($d = 2$ to $8$) and compare the Top-1 accuracy of MPTs (at different pruning rates) with weight-trained dense network. We notice that there exist a range of pruning rates where the performance of MPTs are very similar, and beyond this range the performance drops quickly. Interestingly, as the network depth increases, more parameters can be pruned without hurting the performance of MPTs. For example, MPT-1/32 can match the performance of trained Conv-8 while having only $\sim 20\%$ of its parameter count. Interestingly, the

---

[3]A comparison of MPT-1/32 found using **biprop** and edgepopup is provided in Appendix E, which demonstrates that **biprop** outperforms edgepopup.

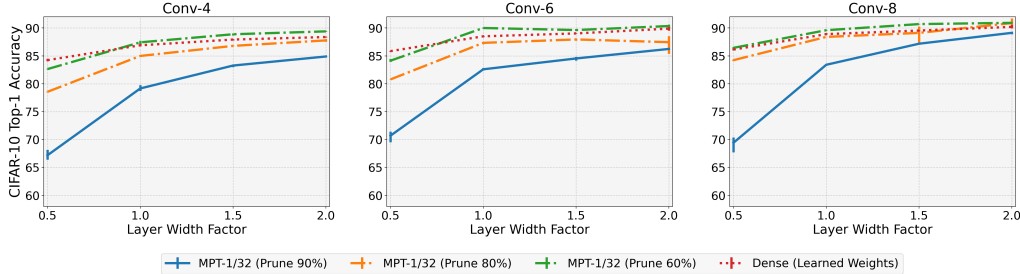

Figure 3: **Effect of Varying Width on MPT-1/32**: Comparing the Top-1 accuracy of sparse and binary MPT-1/32 to dense, full-precision, and weight-optimized network on CIFAR-10.

performance gap between MPT-1/32 and MPT-1/1 does not change much with depth across different pruning rates. We further note that the performance of MPTs improve when increasing the depth and both start to approach the performance of the dense model with learned weights. This gain starts to plateau beyond a certain depth, suggesting that the MPTs might be approaching the limit of their achievable accuracy. Surprisingly, MPT-1/32 performs equally good (or better) than the weight-trained model regardless of having $50 - 80\%$ lesser parameters and weights being binarized.

### 3.1.2 DO WINNING TICKETS EXIST IN WIDE NETWORKS?

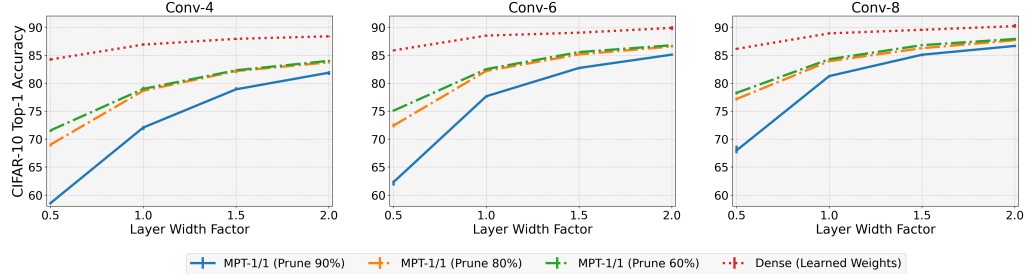

Figure 4: **Effect of Varying Width on MPT-1/1**: Comparing the Top-1 accuracy of sparse and binary MPT-1/1 to dense, full-precision, and weight-optimized network on CIFAR-10.

Similar to the previous experiment, in this experiment, we empirically test the following hypothesis: *As a network grows wider, the performance of multi-prize tickets in the randomly initialized network will approach the performance of the same network with learned weights. We are further interested in exploring the required layer width for our hypothesis to be true.*

In Figures 3 and 4, we vary the width of different VGG architectures and compare the Top-1 accuracy of MPT-1/32 and MPT-1/1 tickets (at different pruning rates) with weight-trained dense network. A width multiplier of value 1 corresponds to the models in Figure 2. Performance of all the models improves when increasing the width and the performance of both MPT-1/32 and MPT-1/1 start to approach the performance of the dense model with learned weights. Although, this gain starts to plateau beyond a certain width. For both MPT-1/32 and MPT-1/1, as the width and depth increase the performance at different pruning rates approach the same value. This observed phenomenon yields a more significant gain in the performance for MPTs with higher pruning rates. Similar to the previous experiment, the performance of MPT-1/32 matches (or exceeds) the performance of dense models for a large range of pruning rates. Furthermore, in the high width regime, a large number of weights ($\sim 90\%$) can be pruned without having a noticeable impact on the performance of MPTs. We also notice that the performance gap between MPT-1/32 and MPT-1/1 decreases significantly with an increase the width which is in sharp contrast with the with the depth experiments where the performance gap between MPT-1/32 and MPT-1/1 appeared to be largely independent of the depth.

**Key Takeaways.** Our experiments verify *Multi-Prize Lottery Ticket Hypothesis* and additionally convey the significance of choosing appropriate network depth and layer width for a given pruning rate. In particular, we find that a network with a large width can be pruned more aggressively without sacrificing much accuracy, while the accuracy of a network with smaller widths suffers when pruning a large percentage of the weights. Similar patterns hold for the depth of the networks as well. The amount of overparametrization needed to approach the performance of dense networks seems to differ for MPT variants – MPT-1/1 requires higher depth and width compared to MPT-1/32.

## 3.2 How Redundant Are State-of-the-Art Deep Neural Networks?

Having shown that MPTs can perform equally good (or better) than overparameterized networks, this experiment aims to answer: *Are state-of-the-art weight-trained DNNs overparametrized enough that significantly smaller multi-prize tickets can match (or beat) their performance?*

**Experimental Configuration.** Instead of focusing on extremely large DNNs, we experiment with small to moderate size DNNs. Specifically, we analyze the redundancy of following backbone models: (1) VGG-Small and ResNet-18 on CIFAR-10, and (2) WideResNet-34 and WideResNet-50 on ImageNet. As we will show later that even these models are highly redundant, thus, our finding automatically extends to larger models. In this process, we also perform a comprehensive comparison of the performance of our multi-prize winning tickets with state-of-the-art in binary neural networks (BNNs). Details on the experimental configuration are provided in Appendix A.

This experiment uses Algorithm 1 to find MPTs within randomly initialized backbone networks. We compare the Top-1 accuracy and number of non-zero parameters for our MPT-1/32 and MPT-1/1 tickets with selected baselines in BNNs (Qin et al., 2020a). Results for CIFAR-10 and ImageNet are shown in Tables 1, 2 and Tables 3, 4, respectively. Next to each MPT method we include the percentage of weights pruned in parentheses. Motivated by (Frankle et al., 2020), we also include models in which the BatchNorm parameters are learned when identifying the random subnetwork using biprop, indicated by +BN. A more comprehensive comparison can be found in Appendix D.

| Method | Model | Top-1 | Params |
|---|---|---|---|
| BinaryConnect | VGG-Small | 91.7 | 4.6 M |
| ProxQuant | ResNet-56 | 92.3 | 0.85 M |
| DSQ | ResNet-20 | 90.2 | 0.27 M |
| IR-Net | ResNet-20 | 90.8 | 0.27 M |
| Full-Precision | ResNet-18 | 93.02 | 11.2 M |
| MPT (80) | ResNet-18 | 94.66 | 2.2 M |
| **MPT (80) +BN** | **ResNet-18** | **94.8** | **2.2 M** |

Table 1: Comparison of MPT-1/32 with trained binary-1/32 networks on CIFAR-10.

| Method | Model | Top-1 | Params |
|---|---|---|---|
| BNN | VGG-Small | 89.9 | 4.6 M |
| XNOR-Net | VGG-Small | 89.8 | 4.6 M |
| DSQ | VGG-Small | 91.7 | 4.6 M |
| IR-Net | ResNet-18 | 91.5 | 11.2 M |
| Full-Precision | VGG-Small | 93.6 | 4.6 M |
| MPT (75) | VGG-Small | 88.52 | 1.44 M |
| **MPT (75) +BN** | **VGG-Small** | **91.9** | **1.44 M** |

Table 2: Comparison of MPT-1/1 with trained binary-1/1 networks on CIFAR-10.

| Method | Model | Top-1 | Params |
|---|---|---|---|
| ABC-Net | ResNet-18 | 62.8 | 11.2 M |
| BWN | ResNet-18 | 60.8 | 11.2 M |
| IR-Net | ResNet-34 | 70.4 | 21.8 M |
| Quant-Net | ResNet-50 | 72.8 | 25.6 M |
| Full-Precision | ResNet-34 | 73.27 | 21.8 M |
| MPT (80) | WRN-50 | 72.67 | 13.7 M |
| **MPT (80) +BN** | **WRN-50** | **74.03** | **13.7 M** |

Table 3: Comparison of MPT-1/32 with trained binary-1/32 networks on ImageNet.

| Method | Model | Top-1 | Params |
|---|---|---|---|
| BNN | AlexNet | 27.9 | 62.3 M |
| XNOR-Net | AlexNet | 44.2 | 62.3 M |
| ABC-Net | ResNet-34 | 52.4 | 21.8 M |
| **IR-Net** | **ResNet-34** | **62.9** | **21.8 M** |
| Full-Precision | ResNet-34 | 73.27 | 21.8 M |
| MPT (60) | WRN-34 | 45.06 | 19.3 M |
| MPT (60) +BN | WRN-34 | 52.07 | 19.3 M |

Table 4: Comparison of MPT-1/1 with trained binary-1/1 networks on ImageNet.

Our results highlight that SOTA DNN models are extremely redundant. For similar parameter count, our binary MPT-1/32 models outperform even full-precision models with learned weights. When compared to state-of-the-art in BNNs, with minimal hyperparameter tuning our multi-prize tickets achieve comparable (or higher) Top-1 accuracy. Specifically, our MPT-1/32 outperform trained

binary weight networks on CIFAR-10 and ImageNet and our MPT-1/1 outperforms trained binary weight and activation networks on CIFAR-10. Further, on CIFAR-10 and ImageNet, MPT-1/32 networks with significantly reduced parameter counts outperform dense and full precision networks with learned weights. Searches for MPT-1/1 in BNN-specific architectures (Kim et al., 2020; Bulat et al., 2020a) and adopting other commonly used tricks to improve model & representation capacities (Bulat et al., 2020b; Yang et al., 2020; Lin et al., 2020; 2021) are likely to yield MPT-1/1 networks with improved performance. For example, up to a 7% gain in the MPT-1/1 accuracy was achieved by simply allowing BatchNorm parameters to be updated. Additionally, alternative approaches for updating the pruning mask in **biprop** could alleviate issues with back-propagating gradients through binary activation networks.

## 4   DISCUSSION AND IMPLICATIONS

Existing compression approaches (e.g., pruning and binarization) typically rely on some form of weight-training. This paper showed that a sufficiently overparametrized randomly weighted network contains binary subnetworks that achieve high accuracy (comparable to dense and full precision original network with learned weights) without any training. We referred to this finding as the *Multi-Prize Lottery Ticket Hypothesis.* We also proved the existence of such winning tickets and presented a generic procedure to find them. Our comparison with state-of-the-art neural networks corroborated our hypothesis. With minimal hyperparameter tuning, our binary weight multi-prize tickets outperformed current state-of-the-art in BNNs and proved its practical importance. Our work has several important practical and theoretical implications.

**Algorithmic.** Our **biprop** framework enjoys certain advantages over traditional weight-optimization. First, contemporary experience suggests that sparse BNN training from scratch is challenging. Both sparseness and binarization bring their own challenges for gradient-based weight training – getting stuck at bad local minima in the sparse regime, incompatibility of back-propagation due to discontinuity in activation function, etc. Although we used gradient-based approaches in this paper, **biprop** is flexible to accommodate different class of algorithms that might avoid the pitfalls of gradient-based weight training. Next, in contrast to weight-optimization that requires large model size and massive compute resources to achieve high performance, our hypothesis suggests that one can achieve similar performance without ever training the large model. Therefore, strategies such as fast ticket search (You et al., 2019) or forward ticket selection (Ye et al., 2020) can be developed to enable more efficient ways of finding–or even designing–MPTs. Finally, as opposed to weight-optimization, **biprop** by design achieves compact yet accurate models.

**Theoretical.** MPTs achieve similar performance as the model with learned weights. First, this observation notes the benefit of overparameterization in the neural network learning and reinforces the idea that an important task of gradient descent (and learning in general) may be to effectively compress overparametrized models to find multi-prize tickets. Next, our results highlight the expressive power of MPTs – since we showed that compressed subnetworks can approximate any target neural network who are known to be universal approximators, our MPTs are also universal approximators. Finally, the multi-prize lottery ticket hypothesis also uncovers the generalization properties of DNNs. Generalization theory for DL is still in its infancy and its not clear what and how DNNs learn (Neyshabur et al., 2017). Multi-prize lottery ticket hypothesis may serve as a valuable tool for answering such questions as it indicates the dependence of generalization on the compressiblity.

**Practical.** Huge storage and heavy computation requirements of state-of-the-art deep neural networks inevitably limit their applications in practice. Multi-prize tickets are significantly lighter, faster, and efficient while maintaining performance. This unlocks a range of potential applications DL could be applied to (e.g., applications with resource-constrained devices such as mobile phones, embedded devices, etc.). Our results also indicate that existing SOTA models might be spending far more compute and power than is needed to achieve a certain performance. In other words, SOTA DL models have terrible energy efficiency and significant carbon footprint (Strubell et al., 2019). In this regard, MPTs have the potential to enable environmentally friendly artificial intelligence.

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

## ACKNOWLEDGEMENTS

The authors would like to thank Shreya Chaganti for her valuable contributions to the **biprop** open source code development and for her help on training MPT models for the final version of the paper.

This work was performed under the auspices of the U.S. Department of Energy by the Lawrence Livermore National Laboratory under Contract No. DE-AC52-07NA27344, Lawrence Livermore National Security, LLC. This document was prepared as an account of the work sponsored by an agency of the United States Government. Neither the United States Government nor Lawrence Livermore National Security, LLC, nor any of their employees makes any warranty, expressed or implied, or assumes any legal liability or responsibility for the accuracy, completeness, or usefulness of any information, apparatus, product, or process disclosed, or represents that its use would not infringe privately owned rights. Reference herein to any specific commercial product, process, or service by trade name, trademark, manufacturer, or otherwise does not necessarily constitute or imply its endorsement, recommendation, or favoring by the United States Government or Lawrence Livermore National Security, LLC. The views and opinions of the authors expressed herein do not necessarily state or reflect those of the United States Government or Lawrence Livermore National Security, LLC, and shall not be used for advertising or product endorsement purposes. This work was supported by LLNL Laboratory Directed Research and Development project 20-ER-014 and released with LLNL tracking number LLNL-CONF-815432.

## A  Hyperparameter Configurations

### A.1  Hyperparameters for Section 3.1

**Experimental Configuration.**  For MPT-1/32 tickets, the network structure is not modified from the original.  For MPT-1/1 tickets, the network structure is modified by moving the max-pooling layer directly after the convolution layer and adding a batch-normalization layer before the binary activation function, as is common in many BNN architectures (Rastegari et al., 2016). We choose our baselines as dense full precision models with learned weights. The baselines were obtained by training backbone networks using the Adam optimizer with learning rate of 0.0003 for 100 epochs and with a batch size of 60. In each randomly weighted backbone network, we find winning tickets MPT-1/32 and MPT-1/1 for different pruning rates using Algorithm 1. For both the weight-optimized and MPT networks, the weights are initialized using the Kaiming Normal distribution (He et al., 2015). All training routines make use of a cosine decay learning rate policy.

| Method | Model | Optimizer | LR | Momentum | Weight Decay | Batch | Epochs |
|---|---|---|---|---|---|---|---|
| MPT-1/32 | Conv2/4/6/8 | SGD | 0.1 | 0.9 | 1e-4 | 128 | 250 |
| MPT-1/1 | Conv2/4/6/8 | Adam | 0.1 | - | 1e-4 | 128 | 250 |

Table 5: Hyperparameter Configurations for CIFAR-10 Experiments

### A.2  Hyperparameters for Section 3.2

In these experiments, the weights are initialized using the Kaiming Normal distribution (He et al., 2015) for all the models except for MPT-1/32 on ImageNet where we use the Signed Constant initialization (Ramanujan et al., 2020) as it yielded slightly better performance. All training routines make use of a cosine decay learning rate policy. For ImageNet training we used a label smoothing value of 0.1 and a learning rate warmup length of 5 epochs.

| Method | Model | Opt. | LR | Momentum | Weight Decay | Batch | Epochs |
|---|---|---|---|---|---|---|---|
| MPT-1/32 | ResNet-18 | SGD | 0.1 | 0.9 | 5e-4 | 256 | 250 |
| MPT-1/32 +BN | ResNet-18 | SGD | 0.1 | 0.9 | 5e-4 | 256 | 250 |
| MPT-1/1 | VGG-Small | Adam | 3.63e-3 | - | 17.335 | 128 | 600 |
| MPT-1/1 +BN | VGG-Small | Adam | 3.63e-3 | - | 1e-4 | 128 | 600 |

Table 6: Hyperparameter Configurations for CIFAR-10 Experiments

| Method | Model | Optimizer | LR | Momentum | Weight Decay | Batch | Epochs |
|---|---|---|---|---|---|---|---|
| MPT-1/32 | WRN-50 | SGD | 0.256 | 0.875 | 3.051757812e-5 | 256 | 120 |
| MPT-1/32 +BN | WRN-50 | SGD | 0.256 | 0.875 | 3.051757812e-5 | 256 | 120 |
| MPT-1/1 | WRN-34 | Adam | 2.56e-4 | - | 3.051757812e-5 | 256 | 250 |
| MPT-1/1 +BN | WRN-34 | Adam | 2.56e-4 | - | 3.051757812e-5 | 256 | 250 |

Table 7: Hyperparameter Configurations for ImageNet Experiments

## B  Existence of Binary-Weight Subnetwork Approximating Target Network

In the following analysis, note that we write $Bin(\{-1, +1\}^{m \times n})$ to denote matrices of dimension $m \times n$ whose components are independently sampled from a binomial distribution with elements $\{-1, +1\}$ and probability $p = 1/2$.

**Lemma 1.** *Let $s \in [d]$, $\alpha \in \left[-\frac{1}{\sqrt{s}}, \frac{1}{\sqrt{s}}\right]$, $i \in [d]$, and $\varepsilon, \delta \geq 0$ be given. Let $\boldsymbol{B} \in \{-1, +1\}^{k \times d}$ be chosen randomly from $Bin(\{-1, 1\}^{k \times d})$ and $\boldsymbol{u} \in \{-1, +1\}^k$ be chosen randomly from*

$Bin(\{-1,+1\}^k)$. *If*

$$k \geq \frac{16}{\varepsilon\sqrt{s}} + 16\log\left(\frac{2}{\delta}\right), \tag{3}$$

*then with probability at least $1-\delta$ there exist masks $\tilde{\boldsymbol{m}} \in \{0,1\}^k$ and $\boldsymbol{M} \in \{0,1\}^{k\times d}$ such that the function $g : \mathbb{R}^d \to \mathbb{R}$ defined by*

$$g(\boldsymbol{x}) = (\tilde{\boldsymbol{m}} \odot \boldsymbol{u})^\intercal \sigma\left(\varepsilon(\boldsymbol{M} \odot \boldsymbol{B})\boldsymbol{x}\right), \tag{4}$$

*satisfies*

$$|g(\boldsymbol{x}) - \alpha x_i| \leq \varepsilon, \tag{5}$$

*for all $\|\boldsymbol{x}\|_\infty \leq 1$. Furthermore, $\|\tilde{\boldsymbol{m}}\|_0 = \|\boldsymbol{M}\|_0 \leq \frac{2}{\varepsilon\sqrt{s}}$, and $\max_{1\leq j\leq k}\|\boldsymbol{M}_{j,:}\|_0 \leq 1$.*

*Proof.* If $|\alpha| \leq \varepsilon$ then taking $\boldsymbol{M} = \boldsymbol{0}$ yields the desired result. Suppose that $|\alpha| > \varepsilon$. Then there exists a $c_i \in \mathbb{N}$ such that

$$c_i\varepsilon \leq |\alpha| \leq (c_i+1)\varepsilon \quad \text{and} \quad |c_i\varepsilon - |\alpha|| \leq \varepsilon. \tag{6}$$

Hence, it follows that

$$|c_i\varepsilon\,\text{sign}(\alpha)x_i - \alpha x_i| = |x_i||c_i\varepsilon - |\alpha|| \leq \varepsilon, \tag{7}$$

where the final inequality follows from (6) and the hypothesis that $\|\boldsymbol{x}\|_\infty \leq 1$. Our goal now is to show that with probability $1-\delta$ the random initialization of $\boldsymbol{u}$ and $\boldsymbol{B}$ yield masks $\tilde{\boldsymbol{m}}$ and $\boldsymbol{M}$ such that $g(\boldsymbol{x}) = c_i\varepsilon\,\text{sign}(\alpha)x_i$.

Now fix $i \in [d]$ and take $k' = \frac{k}{2}$. First, we consider the probability

$$P\left(|\{j \in [k'] : u_j = +1 \text{ and } B_{j,i} = \text{sign}(\alpha)\}| < c_i\right). \tag{8}$$

As $\boldsymbol{u}$ and $\boldsymbol{B}_{:,i}$ are each sampled from a binomial distribution with $k'$ trials, the distribution that the pair $(u_j, B_{j,i})$ is sampled from is a multinomial distribution with four possible events each having a probability of $1/4$. Since we are only interested in the event $(u_j, B_{j,i}) = (+1, \text{sign}(\alpha))$ occurring, we can instead consider a binomial distribution where $P((u_j, B_{j,i}) = (+1, \text{sign}(\alpha)) = \frac{1}{4}$ and $P((u_j, B_{j,i}) \neq (+1, \text{sign}(\alpha)) = \frac{3}{4}$. Hence, using Hoeffding's inequality we have that

$$P\left(|\{j \in [k'] : u_j = +1 \text{ and } B_{j,i} = \text{sign}(\alpha)\}| < c_i\right) \leq \exp\left(-2k'\left(\frac{1}{4} - \frac{c_i}{k'}\right)^2\right) \tag{9}$$

$$= \exp\left(-\frac{1}{8}k' + c_i - 2\frac{c_i^2}{k'}\right) \tag{10}$$

$$< \exp\left(-\frac{1}{8}k' + 2c_i\right), \tag{11}$$

where the final inequality follows since $\exp()$ is an increasing function and $-2\frac{c_i^2}{k'} < 0$. From (6) and the fact that $|\alpha| \leq \frac{1}{\sqrt{s}}$, it follows that

$$c_i \leq \frac{1}{\varepsilon\sqrt{s}}. \tag{12}$$

Combining our hypothesis in (3) with (12) yields that

$$-\frac{1}{8}k' + c_i = -\frac{1}{16}k + c_i \leq -\frac{1}{16}\left(\frac{16}{\varepsilon\sqrt{s}} + 16\log\left(\frac{2}{\delta}\right)\right) + \frac{1}{\varepsilon\sqrt{s}} = \log\left(\frac{\delta}{2}\right). \tag{13}$$

Substituting (13) into (11) yields

$$P\left(|\{j \in [k'] : u_j = +1 \text{ and } B_{j,i} = \text{sign}(\alpha)\}| < c_i\right) < \frac{\delta}{2}. \tag{14}$$

Additionally, it follows from the same argument that

$$P\left(|\{k' < j \le k : u_j = -1 \text{ and } B_{j,i} = -\operatorname{sign}(\alpha)\}| < c_i\right) < \frac{\delta}{2}. \tag{15}$$

From (14) and (15) it follows with probability at least $1 - \delta$ that there exist sets $S_+ := \{j : u_j = +1 \text{ and } B_{j,i} = \operatorname{sign}(\alpha)\}$ and $S_- := \{j : u_j = -1 \text{ and } B_{j,i} = -\operatorname{sign}(\alpha)\}$ satisfying $|S_+| = |S_-| = c_i$ and $S_+ \cap S_- = \emptyset$. Using these sets, we define the components of the mask $\tilde{\boldsymbol{m}}$ and $\boldsymbol{M}$ by

$$\tilde{m}_j = \begin{cases} 1 & : \quad j \in S_+ \cup S_- \\ 0 & : \quad \text{otherwise} \end{cases} \tag{16}$$

and

$$M_{j,\ell} = \begin{cases} 1 & : \quad j \in S_+ \cup S_- \text{ and } \ell = i \\ 0 & : \quad \text{otherwise} \end{cases}. \tag{17}$$

Using the definition of $g(\boldsymbol{x})$ in (4) we now have that

$$g(\boldsymbol{x}) = \sum_{i \in S_+} \sigma\left(\varepsilon \operatorname{sign}(\alpha)x_i\right) - \sum_{i \in S_-} \sigma\left(-\varepsilon \operatorname{sign}(\alpha)x_i\right) \tag{18}$$

$$= c_i \sigma\left(\varepsilon \operatorname{sign}(\alpha)x_i\right) - c_i \sigma\left(-\varepsilon \operatorname{sign}(\alpha)x_i\right) \tag{19}$$

$$= c_i \varepsilon \operatorname{sign}(\alpha)x_i, \tag{20}$$

where the final equality follows from the identity $\sigma(a) - \sigma(-a) = a$, for all $a \in \mathbb{R}$. This concludes the proof of (5).

Lastly, by our choice of $\tilde{\boldsymbol{m}}$ in (16), $\boldsymbol{M}$ in (17), and (12), it follows that

$$\|\tilde{\boldsymbol{m}}\|_0 = \|\boldsymbol{M}\|_0 = 2c_i \le \frac{2}{\varepsilon\sqrt{s}}, \tag{21}$$

and

$$\max_{1 \le j \le k} \|\boldsymbol{M}_{j,:}\|_0 \le 1, \tag{22}$$

which concludes the proof. □

The next step is to consider an analogue for Lemma A.2 from (Malach et al., 2020) which we provide in Lemma 2.

**Lemma 2.** *Let $s \in [d]$, $\boldsymbol{w}^* \in \left[-\frac{1}{\sqrt{s}}, \frac{1}{\sqrt{s}}\right]^d$ with $\|\boldsymbol{w}^*\|_0 \le s$, and $\varepsilon, \delta > 0$ be given. Let $\boldsymbol{B} \in \{-1, +1\}^{k \times d}$ be chosen randomly from $Bin(\{-1, 1\}^{k \times d})$ and $\boldsymbol{u} \in \{-1, +1\}^k$ be chosen randomly from $Bin(\{-1, +1\}^k)$. If*

$$k \ge s \cdot \left\lceil \frac{16\sqrt{s}}{\varepsilon} + 16\log\left(\frac{2s}{\delta}\right) \right\rceil, \tag{23}$$

*then with probability at least $1 - \delta$ there exist masks $\tilde{\boldsymbol{m}} \in \{0,1\}^k$ and $\boldsymbol{M} \in \{0,1\}^{k \times d}$ such that the function $g : \mathbb{R}^d \to \mathbb{R}$ defined by*

$$g(\boldsymbol{x}) = (\tilde{\boldsymbol{m}} \odot \boldsymbol{u})^{\mathsf{T}} \sigma\left(\varepsilon(\boldsymbol{M} \odot \boldsymbol{B})\boldsymbol{x}\right), \tag{24}$$

*satisfies*

$$|g(\boldsymbol{x}) - \langle \boldsymbol{w}^*, \boldsymbol{x} \rangle| \le \varepsilon, \text{ for all } \|\boldsymbol{x}\|_\infty \le 1. \tag{25}$$

*Furthermore, $\|\tilde{\boldsymbol{m}}\|_0 = \|\boldsymbol{M}\|_0 \le \frac{2s\sqrt{s}}{\varepsilon}$ and $\max_{1 \le j \le k} \|\boldsymbol{M}_{j,:}\|_0 \le 1$.*

*Proof.* Assume $k = s \cdot \left\lceil \frac{16\sqrt{s}}{\varepsilon} + 16\log\left(\frac{2s}{\delta}\right) \right\rceil$ and set $k' = \frac{k}{s}$. Note that if $k > s \cdot \left\lceil \frac{16\sqrt{s}}{\varepsilon} + 16\log\left(\frac{2s}{\delta}\right) \right\rceil$ then the excess neurons can be masked yielding the desired value for $k$. We

decompose $\boldsymbol{u}$, $\tilde{\boldsymbol{m}}$, $\boldsymbol{B}$, and $\boldsymbol{M}$ into $s$ equal size submatrices by defining

$$\boldsymbol{u}^{(i)} := \begin{bmatrix} u_{k'(i-1)+1} & \cdots & u_{k'i} \end{bmatrix}^{\mathsf{T}} \in \{-1, +1\}^{k' \times 1} \tag{26}$$

$$\tilde{\boldsymbol{m}}^{(i)} := \begin{bmatrix} \tilde{m}_{k'(i-1)+1} & \cdots & \tilde{m}_{k'i} \end{bmatrix}^{\mathsf{T}} \in \{0, 1\}^{k' \times 1} \tag{27}$$

$$\boldsymbol{B}^{(i)} := \begin{bmatrix} b_{(k'(i-1)+1),1} & \cdots & b_{(k'(i-1)+1),d} \\ \vdots & \ddots & \vdots \\ b_{k'i,1} & \cdots & b_{k'i,d} \end{bmatrix} \in \{-1, +1\}^{k' \times d} \tag{28}$$

$$\boldsymbol{M}^{(i)} := \begin{bmatrix} m_{(k'(i-1)+1),1} & \cdots & m_{(k'(i-1)+1),d} \\ \vdots & \ddots & \vdots \\ m_{k'i,1} & \cdots & m_{k'i,d} \end{bmatrix} \in \{0, 1\}^{k' \times d}, \tag{29}$$

for $i \in [s]$. Note that these submatrices satisfy

$$\boldsymbol{u} = \begin{bmatrix} \boldsymbol{u}^{(1)} \\ \vdots \\ \boldsymbol{u}^{(s)} \end{bmatrix}, \quad \tilde{\boldsymbol{m}} = \begin{bmatrix} \tilde{\boldsymbol{m}}^{(1)} \\ \vdots \\ \tilde{\boldsymbol{m}}^{(s)} \end{bmatrix}, \quad \boldsymbol{B} = \begin{bmatrix} \boldsymbol{B}^{(1)} \\ \vdots \\ \boldsymbol{B}^{(s)} \end{bmatrix}, \quad \boldsymbol{M} = \begin{bmatrix} \boldsymbol{M}^{(1)} \\ \vdots \\ \boldsymbol{M}^{(s)} \end{bmatrix}. \tag{30}$$

Now let $\mathcal{I} := \{i \in [d] : w_i^* \neq 0\}$. By our hypothesis that $\|\boldsymbol{w}^*\|_0 \leq s$, it follows that $|\mathcal{I}| \leq s$. WLOG, assume that $\mathcal{I} \subseteq [s]$. Now fix $i \in [s]$ and define $g_i : \mathbb{R}^d \to \mathbb{R}$ by

$$g_i(\boldsymbol{x}) := \left( \tilde{\boldsymbol{m}}^{(i)} \odot \boldsymbol{u}^{(i)} \right)^{\mathsf{T}} \sigma \left( \varepsilon (\boldsymbol{M}^{(i)} \odot \boldsymbol{B}^{(i)}) \boldsymbol{x} \right) \tag{31}$$

By (23), taking $\varepsilon' = \frac{\varepsilon}{s}$ and $\delta' = \frac{\delta}{s}$ yields that $k' \geq \frac{16}{\varepsilon' \sqrt{s}} + 16 \log \left( \frac{2}{\delta'} \right)$. Hence, it follows from Lemma 1 that with probability at least $1 - \delta'$ there exist $\tilde{\boldsymbol{m}}^{(i)} \in \{0, 1\}^{k'}$ and $\boldsymbol{M}^{(i)} \in \{0, 1\}^{k' \times d}$ such that

$$|g_i(\boldsymbol{x}) - w_i^* x_i| \leq \varepsilon' = \frac{\varepsilon}{s}, \tag{32}$$

for every $\boldsymbol{x} \in \mathbb{R}^d$ with $\|\boldsymbol{x}\|_\infty \leq 1$, and

$$\|\tilde{\boldsymbol{m}}^{(i)}\|_0 = \|\boldsymbol{M}^{(i)}\|_0 \leq \frac{2}{\varepsilon' \sqrt{s}} = \frac{2\sqrt{s}}{\varepsilon} \quad \text{and} \quad \max_{k'(i-1)+1 \leq j \leq k'i} \|M_{j,i}^{(i)}\|_0 \leq 1. \tag{33}$$

By the definition of $g(\boldsymbol{x})$ in (24), using (30) yields

$$g(\boldsymbol{x}) = (\tilde{\boldsymbol{m}} \odot \boldsymbol{u})^{\mathsf{T}} \sigma \left( \varepsilon (\boldsymbol{M} \odot \boldsymbol{B}) \boldsymbol{x} \right) = \sum_{i=1}^s \left( \tilde{\boldsymbol{m}}^{(i)} \odot \boldsymbol{u}^{(i)} \right)^{\mathsf{T}} \sigma \left( \varepsilon (\boldsymbol{M}^{(i)} \odot \boldsymbol{B}^{(i)}) \boldsymbol{x} \right) = \sum_{i=1}^s g_i(\boldsymbol{x}). \tag{34}$$

Hence, combining (32) for all $i \in [s]$, it follows that with probability at least $1 - \delta$ we have

$$|g(\boldsymbol{x}) - \langle \boldsymbol{w}^*, \boldsymbol{x} \rangle| = \left| \sum_{i=1}^s g_i(\boldsymbol{x}) - \sum_{i=1}^s w_i^* x_i \right| \leq \sum_{i=1}^s |g_i(\boldsymbol{x}) - w_i^* x_i| \leq \varepsilon. \tag{35}$$

Finally, it follows from (30) and (33) that

$$\|\tilde{\boldsymbol{m}}\|_0 = \|\boldsymbol{M}\|_0 \leq \frac{2s\sqrt{s}}{\varepsilon} \quad \text{and} \quad \max_{1 \leq j \leq k} \|M_{j,:}\|_0 \leq 1, \tag{36}$$

which concludes the proof. $\square$

We now state and prove an analogue to Lemma A.5 in (Malach et al., 2020) which is the last lemma we will need to establish the desired result.

**Lemma 3.** *Let* $s \in [d]$, $\boldsymbol{W}^* \in \left[-\frac{1}{\sqrt{s}}, \frac{1}{\sqrt{s}}\right]^{n \times d}$ *with* $\|\boldsymbol{W}^*\|_0 \leq s$, $F : \mathbb{R}^d \to \mathbb{R}^n$ *defined by* $F_i(\boldsymbol{x}) = \sigma(\langle \boldsymbol{w}_i^*, \boldsymbol{x} \rangle)$, *and* $\varepsilon, \delta > 0$ *be given. Let* $\boldsymbol{B} \in \{-1, +1\}^{k \times d}$ *be chosen randomly from* $Bin(\{-1, 1\}^{k \times d})$ *and* $\boldsymbol{U} \in \{-1, +1\}^{k \times n}$ *be chosen randomly from* $Bin(\{-1, +1\}^{k \times n})$. *If*

$$k \geq ns \cdot \left\lceil \frac{16\sqrt{ns}}{\varepsilon} + 16 \log\left(\frac{2ns}{\delta}\right) \right\rceil, \tag{37}$$

*then with probability at least* $1 - \delta$ *there exist masks* $\tilde{\boldsymbol{M}} \in \{0, 1\}^{k \times n}$ *and* $\boldsymbol{M} \in \{0, 1\}^{k \times d}$ *such that the function* $G : \mathbb{R}^d \to \mathbb{R}^n$ *defined by*

$$G(\boldsymbol{x}) = \sigma\left( (\tilde{\boldsymbol{M}} \odot \boldsymbol{U})^\mathsf{T} \sigma\left( \varepsilon(\boldsymbol{M} \odot \boldsymbol{B})\boldsymbol{x} \right) \right), \tag{38}$$

*satisfies*

$$\|G(\boldsymbol{x}) - F(\boldsymbol{x})\|_2 \leq \varepsilon, \text{ for all } \|\boldsymbol{x}\|_\infty \leq 1. \tag{39}$$

*Furthermore,* $\|\tilde{\boldsymbol{M}}\|_0 = \|\boldsymbol{M}\|_0 \leq \frac{2ns\sqrt{ns}}{\varepsilon}$.

*Proof.* Assume $k = ns \cdot \left\lceil \frac{16\sqrt{ns}}{\varepsilon} + 16 \log\left(\frac{2ns}{\delta}\right) \right\rceil$ and set $k' = \frac{k}{n}$. Note that if $k > ns \cdot \left\lceil \frac{16\sqrt{ns}}{\varepsilon} + 16 \log\left(\frac{2ns}{\delta}\right) \right\rceil$ then excess neurons can be masked to yield the desired value for $k$. As in the proof of Lemma 2, we can split $\boldsymbol{U}$, $\tilde{\boldsymbol{M}}$, $\boldsymbol{B}$, and $\boldsymbol{M}$ into $n$ submatrices, denoted $\boldsymbol{U}^{(i)} \in \{-1, +1\}^{k' \times n}$, $\tilde{\boldsymbol{M}}^{(i)} \in \{-1, +1\}^{k' \times n}$, $\boldsymbol{B}^{(i)} \in \{-1, +1\}^{k' \times d}$, and $\boldsymbol{M}^{(i)} \in \{-1, +1\}^{k' \times d}$ for $i \in [n]$, such that

$$\boldsymbol{U} = \begin{bmatrix} \boldsymbol{U}^{(1)} \\ \vdots \\ \boldsymbol{U}^{(n)} \end{bmatrix}, \ \tilde{\boldsymbol{M}} = \begin{bmatrix} \tilde{\boldsymbol{M}}^{(1)} \\ \vdots \\ \tilde{\boldsymbol{M}}^{(n)} \end{bmatrix}, \ \boldsymbol{B} = \begin{bmatrix} \boldsymbol{B}^{(1)} \\ \vdots \\ \boldsymbol{B}^{(n)} \end{bmatrix}, \text{ and } \boldsymbol{M} = \begin{bmatrix} \boldsymbol{M}^{(1)} \\ \vdots \\ \boldsymbol{M}^{(n)} \end{bmatrix}. \tag{40}$$

To simplify notation in the following definition, we define the vectors $\tilde{\boldsymbol{m}}^{(i)} := \tilde{\boldsymbol{M}}_{:,i}^{(i)}$ and $\tilde{\boldsymbol{u}}^{(i)} := \tilde{\boldsymbol{U}}_{:,i}^{(i)}$. Now we define the functions $g_i : \mathbb{R}^d \to \mathbb{R}$ by

$$g_i(\boldsymbol{x}) = \left( \tilde{\boldsymbol{m}}^{(i)} \odot \boldsymbol{u}^{(i)} \right)^\mathsf{T} \sigma\left( \beta(\boldsymbol{M}^{(i)} \odot \boldsymbol{B}^{(i)})\boldsymbol{x} \right), \tag{41}$$

for each $i \in [n]$. Taking $\varepsilon' = \frac{\varepsilon}{\sqrt{n}}$ and $\delta' = \frac{\delta}{n}$, it follows from (37) that $k' \geq s \cdot \left\lceil \frac{16\sqrt{s}}{\varepsilon'} + 16 \log\left(\frac{2s}{\delta'}\right) \right\rceil$. As the hypotheses of Lemma 2 are satisfied, with probability at least $1 - \frac{\delta}{n}$ there exist masks $\tilde{\boldsymbol{m}}^{(i)}$ and $\boldsymbol{M}^{(i)}$ with

$$\|\tilde{\boldsymbol{m}}^{(i)}\|_0 = \|\boldsymbol{M}^{(i)}\|_0 \leq \frac{2s\sqrt{s}}{\varepsilon'} = \frac{2s\sqrt{ns}}{\varepsilon} \tag{42}$$

such that

$$|g_i(\boldsymbol{x}) - \langle \boldsymbol{W}_i^*, \boldsymbol{x} \rangle| \leq \frac{\varepsilon}{\sqrt{n}}, \text{ for all } \|\boldsymbol{x}\|_\infty \leq 1. \tag{43}$$

For each $i \in [n]$, note that this results in choosing the columns of the mask $\tilde{\boldsymbol{M}}^{(i)}$ by

$$\tilde{\boldsymbol{M}}_{:,\ell}^{(i)} = \begin{cases} \tilde{\boldsymbol{m}}^{(i)} & : \ \ell = i \\ \boldsymbol{0} & : \ \text{otherwise} \end{cases} \tag{44}$$

Combining this choice with (40) yields

$$(\tilde{\boldsymbol{M}} \odot \boldsymbol{U})^\mathsf{T} \sigma\left( \beta(\boldsymbol{M} \odot \boldsymbol{B})\boldsymbol{x} \right) = \begin{bmatrix} g_1(\boldsymbol{x}) \\ \vdots \\ g_n(\boldsymbol{x}) \end{bmatrix}. \tag{45}$$

By the definition of $G(\boldsymbol{x})$ in (38), it follows from (45) that

$$G(\boldsymbol{x}) = \begin{bmatrix} \sigma(g_1(\boldsymbol{x})) \\ \vdots \\ \sigma(g_n(\boldsymbol{x})) \end{bmatrix}. \tag{46}$$

Combining (43) and (46), we have with probability at least $1 - \delta$ that

$$\|G(\boldsymbol{x}) - F(\boldsymbol{x})\|_2^2 = \sum_{i=1}^{n} \left(\sigma(g_i(\boldsymbol{x})) - \sigma(\langle \boldsymbol{w}_i^*, \boldsymbol{x} \rangle)\right)^2 \leq \sum_{i=1}^{n} \left(g_i(\boldsymbol{x}) - \langle \boldsymbol{W}_i^*, \boldsymbol{x} \rangle\right)^2 \leq \varepsilon^2. \tag{47}$$

Finally, it follows from (42) and (44) that

$$\|\tilde{\boldsymbol{M}}\|_0 = \|\boldsymbol{M}\|_0 \leq \frac{2ns\sqrt{ns}}{\varepsilon} \tag{48}$$

which concludes the proof. $\qquad\square$

We are now ready to prove the main result in Theorem 2.

**Theorem 2.** *Let* $\ell, n, s \in \mathbb{N}$, $\boldsymbol{W}^{(1)*} \in \left[-\frac{1}{\sqrt{s}}, \frac{1}{\sqrt{s}}\right]^{d \times n}$, $\{\boldsymbol{W}^{(i)*}\}_{i=2}^{\ell-1} \in \left[-\frac{1}{\sqrt{n}}, \frac{1}{\sqrt{n}}\right]^{n \times n}$, *and* $\boldsymbol{W}^{(\ell)*} \in \left[-\frac{1}{\sqrt{n}}, \frac{1}{\sqrt{n}}\right]^{1 \times n}$. *Assume that for each* $i \in [\ell]$ *we have* $\|\boldsymbol{W}^{(i)*}\|_2 \leq 1$ *and* $\max_j \|\boldsymbol{W}_j^{(i)*}\|_0 \leq s$. *Define* $F(x) := F^{(\ell)} \circ \cdots \circ F^{(1)}(\boldsymbol{x})$ *where* $F^{(i)}(\boldsymbol{x}) = \sigma(\boldsymbol{W}^{(i)*}\boldsymbol{x})$ *for* $i \in [\ell - 1]$ *and* $F^{(\ell)}(\boldsymbol{x}) = \boldsymbol{W}^{(\ell)*}\boldsymbol{x}$. *Fix* $\varepsilon, \delta \in (0, 1)$.

*Let* $\boldsymbol{B}^{(1)} \in \{-1, +1\}^{k \times d}$ *be sampled from* $Bin(\{-1, +1\}^{k \times d})$, $\{\boldsymbol{B}^{(i)}\}_{i=2}^{\ell} \in \{-1, +1\}^{k \times n}$ *be sampled from* $Bin(\{-1, +1\}^{k \times n})$, $\{\boldsymbol{U}^{(i)}\}_{i=1}^{\ell-1} \in \{-1, +1\}^{k \times n}$ *be sampled from* $Bin(\{-1, +1\}^{k \times n})$ *and* $\boldsymbol{U}^{(\ell)} \in \{-1, +1\}^{k \times 1}$ *sampled from* $Bin(\{-1, +1\}^{k \times 1})$. *If*

$$k \geq ns \cdot \left\lceil \frac{32\ell\sqrt{ns}}{\varepsilon} + 16 \log\left(\frac{2ns\ell}{\delta}\right) \right\rceil, \tag{49}$$

*then with probability at least* $1 - \delta$ *there exist binary masks* $\{\boldsymbol{M}^{(i)}\}_{i=1}^{\ell}$ *and* $\{\tilde{\boldsymbol{M}}^{(i)}\}_{i=1}^{\ell}$ *for* $\{\boldsymbol{B}^{(i)}\}_{i=1}^{\ell}$ *and* $\{\boldsymbol{U}^{(i)}\}_{i=1}^{\ell}$, *respectively, such that the function* $G : \mathbb{R}^d \to \mathbb{R}$ *defined by*

$$G(\boldsymbol{x}) := G^{(\ell)} \circ \cdots \circ G^{(1)}(\boldsymbol{x}), \tag{50}$$

*where*

$$G^{(i)}(\boldsymbol{x}) := \sigma\left((\tilde{\boldsymbol{M}}^{(i)} \odot \boldsymbol{U}^{(i)})^\intercal \sigma(\varepsilon(\boldsymbol{M}^{(i)} \odot \boldsymbol{B}^{(i)})\boldsymbol{x})\right), \text{ for } i \in [\ell - 1] \tag{51}$$

$$G^{(\ell)}(\boldsymbol{x}) := (\tilde{\boldsymbol{M}}^{(i)} \odot \boldsymbol{U}^{(i)})^\intercal \sigma(\varepsilon(\boldsymbol{M}^{(i)} \odot \boldsymbol{B}^{(i)})\boldsymbol{x}), \tag{52}$$

*satisfies*

$$|G(\boldsymbol{x}) - F(\boldsymbol{x})| \leq \varepsilon, \text{ for all } \|\boldsymbol{x}\|_2. \tag{53}$$

*Additionally,* $\|\tilde{\boldsymbol{M}}\|_0 = \|\boldsymbol{M}\|_0 \leq \frac{4ns\ell^2\sqrt{ns}}{\varepsilon}$.

*Proof.* Let $i \in [\ell - 1]$. Using Lemma 3 with $\varepsilon' = \frac{\varepsilon}{2\ell}$ and $\delta' = \frac{\delta}{\ell}$, with probability at least $1 - \frac{\delta}{\ell}$ there exist $\boldsymbol{M}^{(i)}$ and $\tilde{\boldsymbol{M}}^{(i)}$ such that

$$\|G^{(i)}(\boldsymbol{x}) - F^{(i)}(\boldsymbol{x})\|_2 \leq \frac{\varepsilon}{2\ell}, \text{ for all } \|\boldsymbol{x}\|_\infty \leq 1 \tag{54}$$

and

$$\|\tilde{\boldsymbol{M}}^{(i)}\|_0 = \|\boldsymbol{M}^{(i)}\|_0 \leq \frac{2ns\sqrt{ns}}{\varepsilon'} = \frac{4ns\ell\sqrt{ns}}{\varepsilon}. \tag{55}$$

The remainder of the proof follows from applying the same argument as in the proof of Theorem A.6 from (Malach et al., 2020). $\qquad\square$

## C   Motivation for Framework to Identify MPTs

Suppose that $f(\boldsymbol{x}; \boldsymbol{W}^*)$ with optimized weights $\boldsymbol{W}^*$ is a target network that we wish to approximate. Let $g(\boldsymbol{x}; \boldsymbol{W})$ denote the network in which we want to identify a MPT-1/32 that is an $\varepsilon$-approximation of $f(\boldsymbol{x}; \boldsymbol{W}^*)$, for some $\varepsilon > 0$.

Now assume that $g(\boldsymbol{x}; \cdot)$ is Lipschitz continuous with constant $\kappa$, $\boldsymbol{B} \in \{-1, +1\}^m$ are binary parameters for $g$, and $\alpha \in \mathbb{R}$ is gain term. It follows that

$$
\begin{aligned}
\|g\left(\boldsymbol{x}; \alpha(\boldsymbol{M} \odot \boldsymbol{B})\right) - f(\boldsymbol{x}; \boldsymbol{W}^*)\| &\leq \|g\left(\boldsymbol{x}; \alpha(\boldsymbol{M} \odot \boldsymbol{B}) - g(\boldsymbol{x}; \boldsymbol{M} \odot \boldsymbol{B})\| \\
&\quad + \|g(\boldsymbol{x}; \boldsymbol{M} \odot \boldsymbol{W}) - f(\boldsymbol{x}; \boldsymbol{W}^*)\| \\
&< \kappa \|(\boldsymbol{M} \odot \boldsymbol{W}) - \alpha(\boldsymbol{M} \odot \boldsymbol{B})\| \\
&\quad + \|g(\boldsymbol{x}; \boldsymbol{M} \odot \boldsymbol{W}) - f(\boldsymbol{x}; \boldsymbol{W}^*)\|.
\end{aligned}
\tag{56}
$$

If we take $\boldsymbol{M}$ to be a fixed binary mask, we can minimize the error of binarizing the subnetwork parameters $\boldsymbol{M} \odot \boldsymbol{W}$ by solving the optimization problem

$$
\begin{aligned}
\min_{\alpha, \boldsymbol{B}} \quad & \|(\boldsymbol{M} \odot \boldsymbol{W}) - \alpha(\boldsymbol{M} \odot \boldsymbol{B})\|^2 \\
\text{s.t.} \quad & \alpha \in \mathbb{R}, \ \boldsymbol{B} \in \{-1, 1\}^n
\end{aligned}
\tag{57}
$$

where $\boldsymbol{M}$, $\boldsymbol{W}$, and $\boldsymbol{B}$ are stacked into vectors of some length, say $n$. As the pruning mask $\boldsymbol{M}$ is applied to both $\boldsymbol{W}$ and $\boldsymbol{B}$, solving problem (57) is equivalent to solving problem (2) in (Rastegari et al., 2016) with a different dimension. Hence, it immediately follows that one closed form solution for $\boldsymbol{B}$ in problem (57) is

$$
\boldsymbol{B}^* = \text{sign}(\boldsymbol{W}).
\tag{58}
$$

Taking the derivative of the cost function in (57) with respect to $\alpha$ and setting it equal to zero yields

$$
\alpha(\boldsymbol{M} \odot \boldsymbol{B}^*)^\intercal (\boldsymbol{M} \odot \boldsymbol{B}^*) - (\boldsymbol{M} \odot \boldsymbol{W})^\intercal (\boldsymbol{M} \odot \boldsymbol{B}^*) = 0.
\tag{59}
$$

Recalling that $\boldsymbol{M} \in \{0, 1\}^n$ and using (58), we have

$$
(\boldsymbol{M} \odot \boldsymbol{B}^*)^\intercal (\boldsymbol{M} \odot \boldsymbol{B}^*) = \sum_{i=1}^n (M_i B_i^*)^2 = \sum_{i=1}^n M_i^2 (\text{sign}(W_i))^2 = \sum_{i=1}^n M_i = \|\boldsymbol{M}\|_1
\tag{60}
$$

and

$$
(\boldsymbol{M} \odot \boldsymbol{W})^\intercal (\boldsymbol{M} \odot \boldsymbol{B}^*) = \sum_{i=1}^n M_i^2 W_i \, \text{sign}(W_i) = \sum_{i=1}^n M_i |W_i| = \|\boldsymbol{M} \odot \boldsymbol{W}\|_1.
\tag{61}
$$

Substituting (60) and (61) into (59) and solving for $\alpha$ yields the closed form solution

$$
\alpha^* = \frac{\|\boldsymbol{M} \odot \boldsymbol{W}\|_1}{\|\boldsymbol{M}\|_1}.
\tag{62}
$$

Hence, $\alpha^*$ and $\boldsymbol{B}^*$ minimize the right hand side of (56) and, consequently, reduce the approximation error of the MPT-1/32. So when the binarization error, $\|(\boldsymbol{M} \odot \boldsymbol{W}) - \alpha(\boldsymbol{M} \odot \text{sign}(\boldsymbol{W}))\|$, and the subnetwork error, $\|g(\boldsymbol{x}; \boldsymbol{M} \odot \boldsymbol{W}) - f(\boldsymbol{x}; \boldsymbol{W}^*)\|$, are sufficiently small then the binarized subnetwork $g\left(\boldsymbol{x}; \alpha(\boldsymbol{M} \odot \text{sign}(\boldsymbol{W}))\right)$ serves as a good approximation to the target network.

These closed form expressions for the gain term and the binarized weights are the updates used for the gain term and binary subnetwork weights in **biprop** after updating the binary pruning mask.

## D   Comparison of MPTs with binary neural network SOTA

Here we provide a more exhaustive comparison of MPT–1/32 and MPT–1/1 on CIFAR-10 and ImageNet to SOTA methods – BinaryConnect (Courbariaux et al., 2015), BNN (Courbariaux et al., 2016), DoReFa-Net (Zhou et al., 2016), LQ-Nets (Zhang et al., 2018), BWN and XNOR-Net (Rastegari et al., 2016), ABC-Net (Lin et al., 2017), IR-Net (Qin et al., 2020b), LAB (Hou et al., 2016), ProxQuant (Bai et al., 2018), DSQ (Gong et al., 2019), and BBG (Shen et al., 2020). Results for CIFAR-10 can be found in Tables 8 and 9 and results for ImageNet can be found in Tables 10 and 11. Next to the MPT method we include the percentage of weights pruned and the layer width multiplier (if larger than 1) in parentheses.

| Method | Model | Top-1 | Params |
|---|---|---|---|
| BinaryConnect | VGG-Small | 91.7 | 4.6 M |
| BWN | VGG-Small | 90.1 | 4.6 M |
| DoReFa-Net | ResNet-20 | 90.0 | 0.27 M |
| LQ-Nets | ResNet-20 | 90.1 | 0.27 M |
| LAB | VGG-Small | 89.5 | 4.6 M |
| ProxQuant | ResNet-56 | 92.3 | 0.85 M |
| DSQ | ResNet-20 | 90.2 | 0.27 M |
| IR-Net | ResNet-20 | 90.8 | 0.27 M |
| Full-Precision | ResNet-18 | 93.02 | 11.2 M |
| MPT-1/32 (95) | VGG-Small | 91.48 | 0.23 M |
| MPT (80) | ResNet-18 | 94.66 | 2.2 M |
| MPT (80) +BN | **ResNet-18** | **94.8** | **2.2 M** |

Table 8: Comparison of MPT-1/32 with Trained Binary (1/32) Networks on CIFAR-10

| Method | Model | Top-1 | Params |
|---|---|---|---|
| BNN | VGG-Small | 89.9 | 4.6 M |
| XNOR-Net | VGG-Small | 89.8 | 4.6 M |
| DoReFa-Net | ResNet-20 | 79.3 | 0.27 M |
| BBG | ResNet-20 | 85.3 | 0.27 M |
| LAB | VGG-Small | 87.7 | 4.6 M |
| DSQ | VGG-Small | 91.7 | 4.6 M |
| IR-Net | ResNet-18 | 91.5 | 4.6 M |
| Full-Precision | VGG-Small | 93.6 | 4.6 M |
| MPT (75, 1.25x) | VGG-Small | 88.49 | 1.44 M |
| **MPT (75, 1.25x) +BN** | **VGG-Small** | **91.9** | **1.44 M** |

Table 9: Comparison of MPT-1/1 with Trained Binary (1/1) Networks on CIFAR-10

| Method | Model | Top-1 | Params |
|---|---|---|---|
| ABC-Net | ResNet-18 | 62.8 | 11.2 M |
| BWN | ResNet-18 | 60.8 | 11.2 M |
| BWNH | ResNet-18 | 64.3 | 11.2 M |
| PACT | ResNet-18 | 65.8 | 11.2 M |
| IR-Net | ResNet-34 | 70.4 | 21.8 M |
| Quantization-Networks | ResNet-18 | 66.5 | 11.2 M |
| Quantization-Networks | ResNet-50 | 72.8 | 25.6 M |
| Full-Precision | ResNet-34 | 73.27 | 21.8 M |
| MPT (80) | WRN-50 | 72.67 | 13.7 M |
| **MPT (80) +BN** | **WRN-50** | **74.03** | **13.7 M** |

Table 10: Comparison of MPT-1/32 with Trained Binary (1/32) Networks on ImageNet

## E    COMPARISON TO EDGEPOPUP FOR MPT-1/32

Note that binarization step of **biprop** can be avoided while finding MPT-1/32 – by initializing (and pruning) our backbone neural network with binary initialization (e.g., edgepopup with Signed Constant initialization (Ramanujan et al., 2020)). In this specific instance, **biprop** boils down to edgepopup with proper scaling. Next, we compare the performance of MPT-1/32 networks identified using these two approaches. Both networks presented below use the same hyperparameter configurations and are trained for 250 epochs on the CIFAR-10 dataset. We initialize the networks identified with edgepopup using the Signed Constant initialization as it yielded their best performance. MPT-1/32 networks identified using **biprop** are initialized using the Kaiming Normal initialization. We plot the average over three experiments for each pruning percentage and bars extending to the minimum and maximum accuracy for each pruning percentage. Additionally, for each network we include the Top-1 accuracy of a dense model with learned weights. These plots can be found in

| Method | Model | Top-1 | Params |
|---|---|---|---|
| BNN | AlexNet | 27.9 | 62.3 M |
| XNOR-Net | AlexNet | 44.2 | 62.3 M |
| ABC-Net | ResNet-18 | 42.7 | 11.2 M |
| ABC-Net | ResNet-34 | 52.4 | 21.8 M |
| TSQ | AlexNet | 58.0 | 62.3 M |
| WRPN | ResNet-34 | 60.5 | 21.8 M |
| HWGQ | AlexNet | 52.7 | 62.3 M |
| IR-Net | ResNet-18 | 58.1 | 11.2 M |
| **IR-Net** | **ResNet-34** | **62.9** | **21.8 M** |
| Full-Precision | ResNet-34 | 73.27 | 21.8 M |
| MPT (60) | WRN-34 | 45.06 | 19.3 M |
| MPT (60) +BN | WRN-34 | 52.07 | 19.3 M |

Table 11: Comparison of MPT-1/1 with Trained Binary (1/1) Networks on ImageNet

Figure 5. We find that the performance of MPT-1/32 identified with **biprop** outperforms networks identified using edgepopup. This highlights the benefit of binarization (in conjunction with pruning) as a learning strategy.

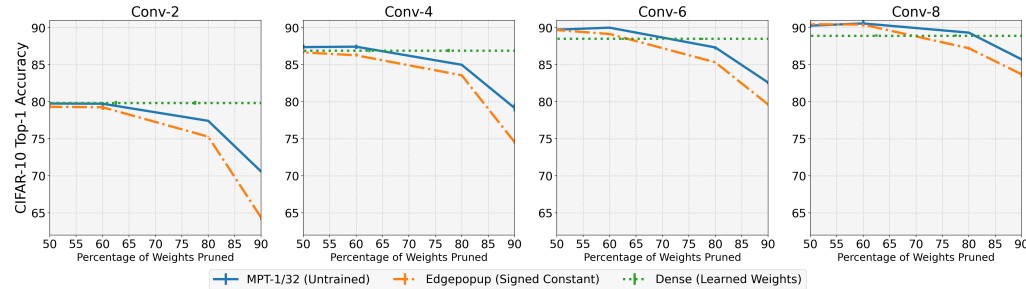

Figure 5: **Comparing biprop and edgepopup**: Comparing the Top-1 accuracy of MPT-1/32 to binary weight networks of the same size identified using edgepopup on CIFAR-10.

## F    RELATED WORK

### F.1    PRUNING

We categorize pruning methods based on whether a model is pruned either after the training or before the training (see (Neill, 2020) for a comprehensive review).

**Post-Training Pruning.**    The traditional pruning methods leverage a three-stage pipeline – pre-training (a large model), pruning, and fine-tuning. The main distinction lies among these approaches is what type of criteria is used for pruning. One of the most popular approach is the magnitude-based pruning where the weights with the magnitude below a certain threshold are discarded (Hagiwara, 1993). Further, certain penalty term (e.g., $l_1$, $l_2$ or lasso weight regularization) can be used during training to encourage a model to learn certain smaller magnitude weights and removing them post-training (Weigend et al., 1991). Models can also be pruned by measuring the importance of weights by computing the sensitivity of the loss function when weights are removed and prune those which cause the smallest change in the loss (LeCun et al., 1990).

**Pruning Before Training.**    Thus far, we have have discussed methods for pruning pretrained DNNs.

Recently, (Frankle & Carbin, 2019) proposed the *Lottery Ticket Hypothesis* and showed that randomly-initialized neural networks contain sparse subnetworks that can be effectively trained from scratch when reset to their initialization. Further, (Liu et al., 2018b) showed that the training

an over-parameterized model is often not necessary to obtain an efficient final model and network architecture itself is more important than the remaining weights after pruning pretrained networks. These findings has revived interest in finding approaches for searching sparse and trainable subnetworks. For example, (Lee et al., 2018; Wang et al., 2020b; You et al., 2019; Wang et al., 2020a) explored efficient approaches to search for these sparse and trainable subnetworks. Along this line of work, a striking finding was reported by (Zhou et al., 2019; Ramanujan et al., 2020) showing that randomly-initialized neural networks contain sparse subnetworks that achieve good performance without any training. (Malach et al., 2020; Pensia et al., 2020) provided theoretical evidences for this phenomenon and showed that one can approximate any target neural network, by pruning a sufficiently over-parameterized network of random weights.

## F.2 BINARIZATION

Similar to pruning, we categorize binarization methods based on whether a model is binarized either after the training or during the training (see (Qin et al., 2020a) for a comprehensive review).

**Post-Training Binarization.**    To the best of our knowledge, none of the post-training schemes have been successful in binarizing pretrained models with or without retraining to achieve reasonable test accuracy. Most existing works (Han et al., 2015; Zhou et al., 2017) are limited to ternary weight quantization.

**Training-Aware Binarization.**    There are several efforts to improve the performance of BNN training. This is a challenging problem as binarization introduces discontinuities which makes differentiation during backpropogation difficult. Binaryconnect (Courbariaux et al., 2015) established how to train networks with binary weights within the familiar back-propagation paradigm. BinaryNet (Courbariaux et al., 2016) further quantize both the weights and the activations to 1-bit values. Unfortunately, these early schemes resulted in a staggering drop in the accuracy compared to their full precision counterparts. In an attempt to improve the performance, XNOR-Net (Rastegari et al., 2016) proposed to add a real-valued channel-wise scaling factor. Dorefa-Net (Zhou et al., 2016) extends XNOR-Net to accelerate the training process using quantized gradients. ABC-Net (Lin et al., 2017) improved the performance by using more weight bases and activation bases at the cost of increase in memory and computation. There have also been efforts in making modifications to the network architectures to make them amenable for the binary neural network training. For example, Bireal-Net (Liu et al., 2018a) added layer-wise identity short-cut, and AutoBNN (Shen et al., 2020) proposed to widen or squeeze the channels in an automatic manner. (Han et al., 2020) proposed to learn to binarize neurons with noisy supervision. Some efforts also have been carried out to designing gradient estimators extending straight-through estimator (STE) (Bengio et al., 2013) for accurate gradient back-propagation. DSQ (Gong et al., 2019) used differentiable soft quantization to have accurate gradients in backward propagation. On the other hand, PCNN Gu et al. (2019) proposed a new discrete back-propagation via projection algorithm to build BNNs.

## F.3 OTHER RELATED DIRECTIONS

Gaier & Ha (2019) proposed a search method for neural network architectures that can already perform a task without any explicit weight training, i.e., each weight in the network has the same shared value. Recent work in randomly wired neural networks (Xie et al., 2019) showed that constructing neural networks with random graph algorithms often outperforms a manually engineered architecture. As opposed to fixed wirings in (Xie et al., 2019), (Wortsman et al., 2019) learned the network parameters as well as the structure. This show that finding a good architecture is akin to finding a sparse subnetwork of the complete graph.

