# OpenReview forum: "Multi-Prize Lottery Ticket Hypothesis: Finding Accurate Binary Neural Networks by Pruning A Randomly Weighted Network"
_ICLR.cc/2021/Conference — ICLR 2021 Poster_

### Official Review · AnonReviewer4 · 2020-10-22
**Good motivation and findings with seem-solid but actually weak theory. unclear paper writing, difficult to follow**

**Rating:** 4
**Confidence:** 4

**Review:**

The paper proposes an innovate method based on lottery ticket hypothesis to prune a BNN (parameters are only -1(0) and +1, it can be viewed as an extreme case of quantization) from a dense NN. It focuses on learning a mask to prune the NN instead of the traditional method (pruning on an already trained network). In addition, not only experiments but theortical proof are given and have a highly brief result.

Pro:
The way to find the mask iteratively is innovate and has a mathematical support.
The result of MPT is amazing because the untrained network can be pruned to a BNN with comparable accrancy of some trained SOTA NN on CIFAR dataset.
The experiments show it can be generized to deeper and wider network.
It has better accurancy than other BNN methods but network parameters still high.
Con:
The main article spends little word to describe how to find the mask, and it is not a trivial way.
The experiments of generization are only done on the very small NN (e.g. Conv2/4/6/8).

Clarity: Very low. Pros: The authors try to express in a way that every step of logical connections in this paper can be clearly understood by readers. To reduce complexity, many parts are settled in the appendix. Cons: Many sentences in this paper are quite long and sometimes using nesting clauses, which makes the text to be obscure. Besides, since many parts are moved into the appendix, the whole structure of the main body is kind of empty and shallow. Some summative and conclusive paragraphs are the simple repetition of previous “claims” since the demonstrations are in the appendix. All of these make a lower clarity.
Finally, I find that this paper has narrower page margins, which means each line can contain more characters. Besides, the header “Under review as a conference paper at …” is missing. These modifications of the submit template may volatile the conference rule and should be considered a cheating behavior. So I give the “very low” score on clarity.

Originality: Medium. Pros: They apply the Lottery Ticket Hypothesis to quantization/BNN. They give proof of their rationality. Cons: The success essentials of their algorithm “biprop” contains two-part: edge-popup and gradient estimator. Both of them are take-away from other works. I regard this work as a new application of LTH to binary neural networks.

---

> ### Author Response · Authors · 2020-11-11
> **“fullpage” package with the ICLR latex template caused formatting issue**
>
> Q. Finally, I find that this paper has narrower page margins, which means each line can contain more characters. Besides, the header “Under review as a conference paper at …” is missing. These modifications of the submit template may volatile the conference rule and should be considered a cheating behavior.
>
> A. We  thank  the  reviewer  for  pointing  this  out.
>
> These  changes  were  not  intentional  but  caused  by mistakenly using the “fullpage” package with the ICLR 21 official latex template and that caused this change.   This  happened when transferring our  draft and used  packages (which included  the“fullpage” package) into the .tex file with the official ICLR 21 latex template.
>
> Further  we  want  to  highlight  that  this  change  has  not  given  us  any  unfair  advantage  –  for  your reference we have uploaded a revised version with the “fullpage” package removed showing that the same information exists in the version with and without this issue.
>
> For these reasons, we hope that the reviewer will not take this issue into account in his/her revised score.
>
> We will respond to your other concerns very soon. Thank you!

---

> ### Author Response · Authors · 2020-11-17
> **Improved writing and additional results have made the paper stronger 1/2**
>
> >The main article spends little word to describe how to find the mask, and it is not a trivial way.
>
> We are sorry for this and thank you for giving us an opportunity to fix this.
>
> In the initial version of the paper, we provided a general pseudocode for the ‘biprop’ framework accommodating a class of potential algorithms. In the revised version, we have instead provided a detailed algorithmic description and theoretical justification of the specific algorithm used in our experimental section (see Section 2.2). Furthermore, we have moved key technical details for finding MPTs from the appendix to the main body.
>
> We hope that these changes have made our algorithm clear. If there is any specific part of the paper that reviewer feels still need more details, we can make that change.
>
>
> > Besides, since many parts are moved into the appendix, the whole structure of the main body is kind of empty and shallow.
>
> In the revised version of the paper, we have made the main text self-contained by bringing back details from the appendix to the main body. This modifications were primarily made to Section 2 in the paper. We would like to emphasize that most of these details were not missing from the paper but were pushed to the appendix to make the paper more readable (at the cost of self-containment). We are grateful to the reviewer for this suggestion as the revised reorganization has made the clarity of the paper significantly better without missing any major details. If there is any specific part of the paper that reviewer feels still need more details, we can make that change.
>
> >The experiments of generization are only done on the very small NN (e.g. Conv2/4/6/8).
> >
>
> We are assuming the question is "Why in Section 3.1 we do not consider large models?" This is done for two main reasons
> 1. Smaller models were faster to train and that gave us an opportunity to perform an in-depth analysis (e.g., varying depth, width, and pruning rate over multiple runs) so that we could obtain reliable empirical patterns
> 2. For our approach, a less overparameterized region is more challenging (as discussed in our theoretical result). Furthermore, if our hypothesis holds for smaller models then it automatically extends to larger models. This is precisely what we showed in Section 3.2 where we considered larger models.
>
> To summarize, experiments on smaller models in not a limitation but the strength of Section 3.1.
>
>
>
> >Many sentences in this paper are quite long and sometimes using nesting clauses, which makes the text to be obscure. Some summative and conclusive paragraphs are the simple repetition of previous “claims” since the demonstrations are in the appendix. All of these make a lower clarity.
>
>
> We have proof-read the whole paper again and tried our best to fix these issues. We would be grateful if reviewer could point us to any such places that we might have missed changing.
>
>
>
>
> ## Sources
>
> [1] Li, Fengfu, Bo Zhang, and Bin Liu. "Ternary weight networks." arXiv preprint arXiv:1605.04711 (2016).
> [2] Wang, Ziheng. "SparseRT: Accelerating Unstructured Sparsity on GPUs for Deep Learning Inference." Proceedings of the ACM International Conference on Parallel Architectures and Compilation Techniques. 2020.
> [3] Ardakani, Arash, Carlo Condo, and Warren J. Gross. "Sparsely-connected neural networks: towards efficient vlsi implementation of deep neural networks." arXiv preprint arXiv:1611.01427 (2016).

---

> > ### Author Response · Authors · 2020-11-17
> > **Improved writing and additional results have made the paper stronger 2/2**
> >
> >
> > >The success essentials of their algorithm “biprop” contains two-part: edge-popup and gradient estimator. Both of them are take-away from other works.
> > >
> >
> > We understand why this confusion might have been caused and it might have appeared that we rely heavily on the edge-popup and the gradient estimator. First, we want to emphasize that the proposed ‘biprop’ framework accommodates a large class of potential algorithms. In our experiments, we used a specific version that had some connections with ‘edge-popup’, but we want to highlight that the only common theme between our implementation of the pruning step and ‘edge-popup’ is that both of us use a common trick of optimizing over binary mask using real-valued score function-based approximation and truncation. This a well-established relaxation technique used in a range of application (see (Joshi & Boyd, 2009)). To further see the difference, note that ‘edge-popup’ cannot accommodate ‘sign’ function to be present in the network. Having ‘sign’ function in MPT-1/32 requires making non-trivial changes such as solving a different optimization problem to approximate a FP NN with binary quantizer and the gain term (see Appendix C and also revised Section 2.2). We derive optimal binary quantizer and the gain term in a closed form. Further, due to binary nature of the activations in MPT-1/1, the gradient vanishes almost everywhere, which is undesirable for the standard back-propagation based score updates for pruning. To overcome this challenge, we use a gradient estimator in the backward pass as pointed out by the reviewer. During the rebuttal period, we realized a novel interpretation of the gradient estimator using quadratic spline parameterization – this interpretation may be of an independent interest to the BNN community. Using this, we show that existing gradient estimator in (Liu et al., 2018a) is a special case of the discussed method. Furthermore, our implementation of the gradient estimator is 6x more memory efficient than the implementation provided in the code used by the authors of (Liu et al., 2018a). Note that all of these issues do not appear in ‘edge-popup’ and further our approach is not a trivial plug-and-play between ‘edge-popup’ and gradient estimation technique.
> >
> > We have highlighted this in the revised version of the paper and explained our algorithm in detail.
> >
> >
> >
> > >Originality: Medium. They apply the Lottery Ticket Hypothesis to quantization/BNN. I regard this work as a new application of LTH to binary neural networks.
> > >
> >
> > We respectfully disagree with the reviewer on this. Our work is not trivially applying existing lottery ticket hypothesis (Frankle 2018) to BNNs. Our hypothesis significantly generalizes existing results, and our algorithms and theory are entirely different.
> >
> > We have made original contributions on three fronts:
> > 1. Proposed generalization of (or much stronger) lottery ticket hypothesis -- highlights the extent of the DNN compression and a new paradigm to learn compact yet highly accurate binary neural nets,
> > 2. Developed high performing algorithms to find these winning tickets – our MPT-1/32 are current SOTA for BWNs and also beat their large and FP counterparts without any significant hyperparameter tuning or commonly used tricks for BNNs, and
> > 3. Presenting first theoretical result that pruned BNNs are universal approximators -- proving that pruning a randomly initialized binary-weight DNN can approximate a real-valued target DNN.
> >
> > Further, these advances are not a trivial extension of the related full-precision literature.
> >
> >
> >
> > >Weak theory, unclear paper writing, difficult to follow
> > >
> > We are sorry for this and thank you for giving us an opportunity to fix this.
> > 1. In the initial version, we only provided an informal statement of our main theoretical result in the main body to improve the readability. In the revised version, we have provided more specific details on the result and also provided a proof sketch in the main body of the paper.
> > 2. In the initial version of the paper, we provided a general pseudocode for the ‘biprop’ framework accommodating a class of potential algorithms. In the revised version, we have instead provided a detailed algorithmic description and theoretical justification of the specific algorithm used in our experimental section.
> >
> > We hope that these changes have made theory and algorithm clear. We would be happy to make any further changes suggested by the reviewer.
> >
> >
> > We would further like to point you to our joint response to all the reviewers to highlight novelty, revised results, and most importantly usefulness and implications of our approach on BNNs and DNNs in general.
> >
> > We hope that the reviewer would agree that the comparison is fair, and our results are much stronger than initially perceived. We would be happy to clarify any further questions/concerns the reviewer might have.

---

> ### Author Response · Authors · 2020-11-23
> **Follow-up on rebuttal**
>
> Following our rebuttal and discussion with rest of the reviewers, we hope that you will find your main concerns addressed, and you will also champion our paper for acceptance. If you have any remaining concerns or questions, please let us know before the discussion period ends so that we can address them.

---

### Official Review · AnonReviewer3 · 2020-10-28
**lottery mask + binary = Ternary ?**

**Rating:** 7
**Confidence:** 3

**Review:**

This work investigated a method of finding a subnetwork of redundant binary networks to gain an overall advantage over pruned or quantized networks.

One main concern of the reviewer is the similarity between the paper's approach---training a mask over a binary network---and the conventional ternary network.  Are the masked weights analogous to the 0 of the ternary networks, while the unmasked weights are in {-1, 1}? In that case, it's fairer to compare with ternary networks with 0 counted into the total params.

Following this analogy, it is also misleading to claim that the network is "untrained", as to minimize the loss, any binarized weights can be updated to 0(masked), although indeed the weight update across 0 is forbidden.

The theory works of this paper are strong and prove that the expressive power of redundant binary(or ternary?) networks can match their denser counterpart.  Still, the question to clarify here is that whether "subset (lottery ticket) + binary network" equals to "ternary network".

In general, the paper is well written and the theoretical and experimental works support the authors' claim. The reviewer would recommend accepting the paper on the condition that the authors can address the comparison with the ternary network fairly.

---

> ### Author Response · Authors · 2020-11-17
> **We have clarified the difference with ternary network**
>
> >Are the masked weights analogous to the 0 of the ternary networks, while the unmasked weights are in {-1, 1}? In that case, it's fairer to compare with ternary networks with 0 counted into the total params.
>
> Zero is not counted into the total parameters to highlight the amount of sparsity in the obtained networks. If we count '0' towards the total parameter count, the number of total parameters for all networks will be the same and this will be unable to highlight the performance of the pruning. Reporting sparsity is a well-accepted practice/metric in the pruning literature -- in fact the goal of  pruning is precisely to increase the amount of sparsity (weight values equal to ‘0’) without dropping the performance. The reason being that having sparser NNs may enable the application of sparse data structures for memory and compute efficiency. To clarify this, in the revised manuscript, we mention “non-zero” parameters when reporting these numbers.
>
> >Following this analogy, it is also misleading to claim that the network is "untrained", as to minimize the loss, any binarized weights can be updated to 0(masked), although indeed the weight update across 0 is forbidden.
> >
>
> The reason for using the term was to differentiate our approach from conventional approaches that optimize over the weights as we are only dropping/pruning connections from a randomly initialized/weighted binary neural network. The reviewer correctly pointed out that we indeed optimize over the mask (‘0’ values). To make our usage clear, we have clarified the term “untrained” in the revised manuscript. However, we are open to completely removing this term if reviewer suggests so.
>
> >The theory works of this paper are strong ...redundant binary (or ternary?) networks can match their denser counterpart. Still, the question to clarify here is that whether "subset (lottery ticket) + binary network" equals to "ternary network".
>
> Our work proves the expressive power of pruning randomly initialized/weight binary neural networks. We would prefer using the term pruned randomly initialized binary neural networks to avoid confusing our approach with ternary neural network approaches in the literature [1]. Comparison with TNN is discussed next.
>
> >The reviewer would recommend accepting the paper on the condition that the authors can address the comparison with the ternary network fairly.
>
> Thank you for asking and giving us an opportunity to answer this excellent question.
> Note that our approach does not fall under any existing framework including binary and ternary neural networks learned via weight-optimization. One could claim that our setting is more restrictive than even binary neural networks/BNN (let alone ternary neural networks) as we learn by pruning “randomly weighted” binary neural network. In other words, unlike existing BNN approaches, we do not optimize over weights and learn purely by pruning – pruning mask is the only optimizable parameter. As we show in the paper, our approach can match (or beat) even full precision neural networks trained using weight-optimization for all considered cases except MPT-1/1 on ImageNet. Thus, they match (or beat) any M-ary quantization including ternary neural networks.
>
> The reviewer is correct in pointing out that weights in the original/large structure can be seen as taking ternary values as pruned weights can be consider having the value ‘0’. But note the activations still take binary values not ternary in MPT-1/1. We differ from ternary weight networks (TWNs) [1] due to the following reasons: 1) In addition to unpruned weights in MPTs (i.e., {+1,-1}) not being trained, we allow a precise control over the sparsity, i.e., number of weights taking the value ‘0’, and 2) As our goal is pruning, our solutions are significantly sparser than TWNs – TWNs achieve their best solutions with sparsity < 50% (usually 20%-40%), and our solutions are always with sparsity >50% (usually 60%-95%). This enables higher energy efficiency and more memory saving (sometimes even when compared to BNNs).
>
> Informally, we could say that (a) in terms of memory and compute requirements: MPT $\leq$ BNN < TNN < FPN, and (b) in terms of performance/accuracy: MPT $\geq$ FPN > TNN > BNN (for MPT-1/32).
>
> We would further like to point you to our joint response to all the reviewers to highlight novelty, revised results, and most importantly usefulness and implications of our approach on BNNs and DNNs in general. We hope that the reviewer would agree that the comparison is fair, and our results are much stronger than initially perceived. We would be happy to clarify any further questions/concerns the reviewer might have.
>
> [1] Ternary weight networks. arXiv:1605.04711 (2016).
> [2] SparseRT: Accelerating Unstructured Sparsity on GPUs for Deep Learning Inference.  https://arxiv.org/abs/2008.11849
> [3] Sparsely-connected neural networks: towards efficient vlsi implementation of deep neural networks. arXiv:1611.01427 (2016).

---

> > ### Comment · AnonReviewer3 · 2020-11-18
> > **good point on that activations remain 2b without zeros.**
> >
> > Thanks for answering my questions.
> > I missed the good point that the activation remains binary when there's "0" in weights. For that, I would like to increase the score to 7.
> > For the hardware with logic to skip zeros, indeed, more sparsity could help. However, from the point of view of hardware designers, non-structured sparsity has limited merit.
> > On the expressiveness, to me, masked numbers ("0") still increase the expressiveness by being at the right place and doing nothing.

---

### Official Review · AnonReviewer1 · 2020-10-29
**Review Paper 738**

**Rating:** 7
**Confidence:** 3

**Review:**

The authors propose a stronger lottery ticket hypothesis in this paper – the multi-prize lottery ticket hypothesis. In particular, the new hypothesis seeks answer to the required amount of over-parameterization for a randomly initialized network to become able to compress to a sparse untrained binary subnetwork with on-par accuracy. The authors prove the existence of such subnetwork and show the bounds on over-parameterization. The paper proposes new methods to get the binary-weight tickets and the binary-activation tickets, where binary-weight tickets are subnetworks with weights as binary, and binary-activation tickets have the activation function in the forward propagation as binary. As binary networks can largely reduce the computational complexity for inference, this work has practical importance especially for applications with constraints for memory and power. The paper has many simulation results to support the theoretical guarantees, and the proposed approach on binary-weight networks has advantages over existing methods.

The paper has sufficient and novel contributions, both for the theoretical results on binary subnetworks and the empirical evaluations that reveal the efficiency of the algorithm on binary-weight subnetworks, so that I recommend this paper for publication.

Here are some minor concerns.

[a] It would be more convincing if the authors could further highlight the technical novelty of this work for the proof of Theorem 1. Compared with Malach 2020 “Proving the lottery ticket hypothesis: pruning is all you need”, the authors can highlight what key differences are needed for proof of the binary network.

[b] For the comparison of full precision network and the MPT from this paper, it can be useful if the computational complexity is shown, where the complexity for MPT would be for the algorithm to find the mutli-prize tickets in an over-parameterized network.

[c] The MPT 1/1 seems not to perform as well compared to trained binary activation network, which may reduce the quality of paper. It is good that the authors mention future work for MPT 1/1 network to attain state-of-art results of binary activation network.

[d] Some of the references seem to lack information, e.g. Malach 2020, Orseau 2020, both of which do not have the venue or journal names.

[e] A question: for the binary weights subnetworks, what does it mean by 20 million parameters? The weights can only be -1 or 1, so the network has 20 million of -1 or 1, but they do not require any multiplication at the inference stage?

---

> ### Author Response · Authors · 2020-11-17
> **We have addressed the minor concerns 1/2**
>
> Thank you for your positive review!
>
> >[a] It would be more convincing if the authors could further highlight the technical novelty of this work for the proof of Theorem 1. Compared with Malach 2020 “Proving the lottery ticket hypothesis: pruning is all you need”, the authors can highlight what key differences are needed for proof of the binary network.
>
> To the best of our knowledge ours is the first theoretical result proving that pruning a randomly initialized binary-weight DNN can approximate a real-valued target DNN. As it has been established that real-valued DNNs are universal approximators (Scarselli & Tsoi, 1998), our result carries the implication that pruned binary-weight DNNs are universal approximators as well. In relation to the first result establishing the existence of real-valued subnetworks in a randomly weighted DNN approximating a real-valued target DNN (Malach et al., 2020), the lower bound on the width established in Theorem 2 is better than their lower bound of $O\left( \ell^2 n^2 \log(\ell n / \delta) /\varepsilon^2 \right)$. While the methodology of our proof is similar to (Malach et al., 2020), proving our result for binary-weight subnetworks is not a trivial extension of their technique and require different tools. The use of binary weights in the analysis results in more complex scenarios that have to be addressed carefully. In particular, the analysis in Lemma 1 requires cleverly splitting a complex scenario involving a multinomial distribution into two simpler cases involving binomial distributions. Then, Hoeffding's inequality is used to derive probabilistic bounds for in each case. The results are combined to yield the final probabilistic bound. This analysis is more complex than the analysis required for real-valued subnetworks as Hoeffding's inequality was not required to establish an analogous result for real-valued subnetworks (Malach et al., 2020). Additionally, the use of binary weights required us to identify an appropriate gain term for rescaling the binary weights in our analysis. Binary weights also result in potentially larger subnetworks being required to approximate a target network and we must carefully keep track of these requirements as we establish results for increasingly larger networks in Lemma 2, Lemma 3, and Theorem 2 in Appendix B. In the work on real-valued networks (Malach et al., 2020) the bound on the required number of parameters in the subnetwork is more straightforward. Furthermore, our analysis yields a tighter bound on the width of the binary network than the bound established in (Malach et al., 2020) for real-valued subnetworks. We hope that this insight into some of the complexity required when analyzing binary subnetworks in our paper illustrates how the analysis is more complex when working with binary subnetworks.
>
> >[b] For the comparison of full precision network and the MPT from this paper, it can be useful if the computational complexity is shown, where the complexity for MPT would be for the algorithm to find the mutli-prize tickets in an over-parameterized network.
>
> For both MPTs and FP weight-trained baselines, we used the same number of epochs as mentioned in the Appendix A.1. Theoretically, MPTs can be trained more efficiently than their FP counterparts. However, achieving this would require their implementation on specialized hardwares, e.g., FPGA, and, this is out of the scope of our paper.

---

> > ### Author Response · Authors · 2020-11-17
> > **We have addressed the minor concerns 2/2**
> >
> >
> > >[c] The MPT 1/1 seems not to perform as well compared to trained binary activation network, which may reduce the quality of paper. It is good that the authors mention future work for MPT 1/1 network to attain state-of-art results of binary activation network.
> >
> >
> > Note that the main contribution of this work is to provide a new paradigm for learning compact yet accurate binary neural networks by pruning and quantizing randomly weighted full precision DNNs. The only case where MPT-1/1 performance is not comparable to the performance of BNN SOTA is ImageNet dataset.
> >
> > Note we have quickly tuned hyperparameters for the MPT-1/1 on CIFAR-10. This has resulted in a boost in accuracy from 88.22\% to 89.59\% and a reduction in the parameter count from 3.68M to 1.61M. The best performing methods DSQ with VGG-small achieves 91.7\% accuracy, however at the cost of (a) almost three-times more parameters, and (b) a more complex quantization scheme. We are confident that some of the tricks used in the BNN literature can be used in the MPT setup as well and will in turn make MPT-1/1 perform better than SOTA BNNs.
> >
> >
> > >[d] Some of the references seem to lack information, e.g. Malach 2020, Orseau 2020, both of which do not have the venue or journal names.
> >
> > Thank you for pointing this out. We have fixed this issue in the revised version.
> >
> > >[e] A question: for the binary weights subnetworks, what does it mean by 20 million parameters? The weights can only be -1 or 1, so the network has 20 million of -1 or 1, but they do not require any multiplication at the inference stage?
> >
> > The reviewer's interpretation is correct – it would imply having have 20 million parameters (or weights/connections) that have values either -1 or +1. Note that due to the binarization, the heavy matrix multiplication operations (MAC) can be replaced with light-weight bitwise XNOR operations and Bitcount operations.
> >
> >
> > We would further like to point you to our joint response to all the reviewers to highlight novelty, revised results, and most importantly usefulness and implications of our approach on BNNs and DNNs in general.
> >
> > We hope that the reviewer would agree that the comparison is fair, and our results are much stronger than initially perceived. We would be happy to clarify any further questions/concerns the reviewer might have.
> >
> > ## Sources
> >
> > [1] Li, Fengfu, Bo Zhang, and Bin Liu. "Ternary weight networks." arXiv preprint arXiv:1605.04711 (2016).
> > [2] Wang, Ziheng. "SparseRT: Accelerating Unstructured Sparsity on GPUs for Deep Learning Inference." Proceedings of the ACM International Conference on Parallel Architectures and Compilation Techniques. 2020.
> > [3] Ardakani, Arash, Carlo Condo, and Warren J. Gross. "Sparsely-connected neural networks: towards efficient vlsi implementation of deep neural networks." arXiv preprint arXiv:1611.01427 (2016).

---

### Official Review · AnonReviewer2 · 2020-10-30
**This paper has some good results, but the writing needs to be improved**

**Rating:** 6
**Confidence:** 3

**Review:**

This paper propose utilizing the existing "lottery ticket" result for constructing binary neural networks. This work has some novelty, in the sense that I haven't seen any other papers on untrained binary neural networks. The experimental results looks good.

However, I have some concerns on this paper.
1. This paper, at least the main text, is not self-contained. The writing needs significant improvement. The main contribution of the paper, section 2, is only one-page long. Neither the theory nor the algorithm are well explained in the main text. Moreover, the algorithm relies heavily on the edge-popup algorithm, which is not explained even in the supplementary material. The title is also somewhat too long.
2. Though the proposed algorithm achieves excellent results in terms of parameter count and accuracy, I think the comparison is somewhat unfair. The subnetwork is sparse, and can be much slower on real hardware. Moreover, pruning from a larger network is known to achieve better result than training a smaller network from scratch, but the baselines does not utilize this.
3. There lacks any discussion on the real time consumption of the proposed network.
4. Time consumption of training should also be reported.

Post rebuttal
====

Thanks the authors for clarifying and revising the paper. The updated version does look much clearer to me, so I updated my ratings.

I am still wondering the difference of biprop vs. a classical quantization-aware training for ternary networks. I did read the response to R3. From my understanding it seems that:
1. biprop doesn't count 0 as a parameter, while TWN does;
2. biprop prunes a larger network (WRN50), while TWN trains a network of the original size (ResNet-50);
I am not sure if the superiority of biprop comes from these reasons, instead of LTH itself. biprop still looks more like a QAT algorithm than a LT-finding algorithm in the sence that
1. it does not train the pruned network after finding the LT as the original LTH paper;
2. it directly learns the binary weights.
Just out of curiosity, but I think clarifying these concerns would make the paper stronger.

---

> ### Author Response · Authors · 2020-11-17
> **Improved writing and additional results have made the paper stronger 1/3**
>
> >However, I have some concerns on this paper. This paper, at least the main text, is not self-contained. The writing needs significant improvement.
>
> In the revised version of the paper, we have made the main text self-contained by bringing back details from the appendix to the main body. We would like to emphasize that most of these details were not missing from the paper but were pushed to the appendix to make the paper more readable (at the cost of self-containment). We are grateful to the reviewer for this suggestion as the revised reorganization has made the clarity of the paper significantly better without missing any major details. If there is any specific part of the paper that reviewer feels still need more details, we can make that change.
>
>
> >The main contribution of the paper, section 2, is only one-page long.
>
> We would like to bring the attention of the reviewer to the fact that the challenge we faced was not that we did not have sufficient technical results but that we made significantly more technical contribution than we could fit in the main body of the paper. Note that related full-precision NN papers are either empirical (Frankle 2018 and Zhou 2019), algorithmic (Ramanujan 2020) or theoretical (Malach 2020). We have made notable progress on all three fronts:
> 1. proposed generalization of (or much stronger) lottery ticket hypothesis -- highlights the extent of the DNN compression and a new paradigm to learn binary neural nets,
> 2. developed high performing algorithms to find these winning tickets – our MPT-1/32 are current SOTA for BWNs and also beat their large and FP counterparts without any significant hyperparameter tuning or commonly used tricks for BNNs, and
> 3. presenting first theoretical result that pruned BNNs are universal approximators -- proving that pruning a randomly initialized binary-weight DNN can approximate a real-valued target DNN.
>
> Further, these advances are not a trivial extension of the related full-precision literature (explained later).
>
> >Though the proposed algorithm achieves excellent results in terms of parameter count and accuracy, I think the comparison is somewhat unfair. The subnetwork is sparse and can be much slower on real hardware.
>
> We respectfully disagree with the comment that “the comparison is unfair as subnetwork is sparse and can be much slower on real hardware”. This comment is not specific our approach but to any pruning-based approach. We think what reviewer is trying to say is that sparse NNs cannot achieve meaningful speedup on commodity hardware (e.g., GPU) built for dense matrix computations. This is indeed true and this is the precise reason that specialized accelerators are designed to exploit the sparsity. However, this does not refute our claims or make the comparison unfair in any way. The sparse model has fewer parameters and, theoretically, less computation costs and approaches such as [2,3] can be leveraged to achieve significant speed ups. However, the hardware implementation of MPTs is out of the scope of this paper and is mentioned as a worthwhile future direction in the revised manuscript.
>
> >Neither the theory nor the algorithm are well explained in the main text.
>
> We are sorry for this and thank you for giving us an opportunity to fix this.
> 1. In the initial version, we only provided an informal statement of our main theoretical result in the main body to improve the readability. In the revised version, we have provided more specific details on the result and also provided a proof sketch in the main body of the paper.
> 2. In the initial version of the paper, we provided a general pseudocode for the ‘biprop’ framework accommodating a class of potential algorithms. In the revised version, we have instead provided a detailed algorithmic description and theoretical justification of the specific algorithm used in our experimental section.
>
>
>
>
> We hope that these changes have made theory and algorithm clear. We would be happy to make any further changes suggested by the reviewer.

---

> > ### Author Response · Authors · 2020-11-17
> > **Improved writing and additional results have made the paper stronger 2/3**
> >
> > >Moreover, the algorithm relies heavily on the edge-popup algorithm, which is not explained even in the supplementary material.
> >
> >
> > We understand why this confusion might have been caused and it might have appeared that we rely heavily on the edge-popup. First, we want to emphasize that the implementation of ‘biprop’ is motivated by a more general framework accommodates a large class of potential algorithms. In our experiments, we used a specific version that had some connections with ‘edge-popup’, but we want to highlight that the only common theme between our implementation of the pruning step and ‘edge-popup’ is that both of us use a common trick of optimizing over binary mask using real-valued score function-based approximation and truncation. This a well-established relaxation technique used in a range of application (see (Joshi & Boyd, 2009)).
> > To further see the difference, note that ‘edge-popup’ cannot accommodate ‘sign’ function to be present in the network. Having ‘sign’ function in MPT-1/32 requires making non-trivial changes such as solving a different optimization problem to approximate a FP NN with binary quantizer and the gain term (see Appendix C and also revised Section 2.2). We derive optimal binary quantizer and the gain term in a closed form. Further, due to binary nature of the activations in MPT-1/1, the gradient vanishes almost everywhere, which is undesirable for the standard back-propagation based score updates for pruning. To overcome this challenge, we use a gradient estimator in the backward pass. During the rebuttal period, we realized a novel interpretation of the gradient estimator using quadratic spline parameterization – this interpretation may be of an independent interest to the BNN community. Using this, we show that gradient estimator in (Liu et al., 2018a) is a special case of the discussed method. Note that, all of these issues do not appear in ‘edge-popup’. Furthermore, our implementation of the gradient estimator is 6x more memory efficient than the implementation provided in the code used by the authors of (Liu et al., 2018a).
> >
> > We have highlighted this in the revised version of the paper and explained our algorithm in detail.
> >
> >
> > >Moreover, pruning from a larger network is known to achieve better result than training a smaller network from scratch, but the baselines does not utilize this.
> >
> > Note that the baselines we use are weight-trained full precision NNs with the same architecture that is randomly initialized to search MPTs. In other words, we use the strongest baseline that is weight-trained, full precision, LARGE/original neural net and not a smaller network as pointed out by the reviewer. As we show in the paper that our approach is comparable to (or beat) even full precision neural networks trained using weight-optimization for all considered cases except MPT-1/1 on ImageNet. Thus, they match (or beat) any M-ary quantization including ternary (or pruned binary) neural networks trained using weight-optimization. This further emphasizes the impressiveness of our results.
> >
> > Also please see our response to Reviewer 3 on the comparison with ternary NN.
> >
> >
> >
> >
> > >There lacks any discussion on the real time consumption of the proposed network.Time consumption of training should also be reported.
> >
> > For both MPTs and FP weight-train baselines, we used same number of epochs as mentioned in the Appendix A.1. Theoretically, MPTs can be trained more efficiently than their FP counterparts. However, achieving this and a fair train time comparison would require their implementation on the specialized hardware, e.g., FPGA and, this is out of the scope of our paper. Although minimizing training time was not our focus, we have mentioned in Section 4 (Algorithmic) addition ways to accelerate the training. Informally, we could say that MPTs require less time to train than both FP and quantized NNs.

---

> > > ### Author Response · Authors · 2020-11-17
> > > **Improved writing and additional results have made the paper stronger 3/3**
> > >
> > > > This paper propose utilizing the existing "lottery ticket" result for constructing binary neural networks.
> > >
> > > We respectfully disagree with the reviewer on this and would like to clarify. Our work does not utilize the existing lottery ticket hypothesis (Frankle 2018) to construct/train BNNs. Our hypothesis significantly generalizes existing results, and our algorithms and theory are entirely different.
> > >
> > > We have made original contributions on three fronts:
> > > 1. Proposed generalization of (or much stronger) lottery ticket hypothesis -- highlights the extent of the DNN compression and a new paradigm to learn compact yet highly accurate binary neural nets,
> > > 2. Developed high performing algorithms to find these winning tickets – our MPT-1/32 are current SOTA for BWNs and also beat their large and FP counterparts without any significant hyperparameter tuning or commonly used tricks for BNNs, and
> > > 3. Presenting first theoretical result that pruned BNNs are universal approximators -- proving that pruning a randomly initialized binary-weight DNN can approximate a real-valued target DNN.
> > >
> > > Hence, these advances are not a straightforward utilization of the related full-precision literature.
> > >
> > > We would further like to point you to our joint response to all the reviewers to highlight novelty, revised results, and most importantly usefulness and implications of our approach on BNNs and DNNs in general.
> > >
> > > We hope that the reviewer would agree that the comparison is fair, and our results are much stronger than initially perceived. We would be happy to clarify any further questions/concerns the reviewer might have.
> > >
> > >
> > > ## Sources
> > >
> > > [1] Li, Fengfu, Bo Zhang, and Bin Liu. "Ternary weight networks." arXiv preprint arXiv:1605.04711 (2016).
> > > [2] Wang, Ziheng. "SparseRT: Accelerating Unstructured Sparsity on GPUs for Deep Learning Inference." Proceedings of the ACM International Conference on Parallel Architectures and Compilation Techniques. 2020.
> > > [3] Ardakani, Arash, Carlo Condo, and Warren J. Gross. "Sparsely-connected neural networks: towards efficient vlsi implementation of deep neural networks." arXiv preprint arXiv:1611.01427 (2016).

---

> > ### Comment · AnonReviewer2 · 2020-11-23
> > **Algorithm 1**
> >
> > Thanks the authors for the clarification. I am still not entirely clear about Alg. 1.
> >
> > - In Alg. 1, the only loop is the epochs. There are no loop over minibatches. Is biprop a batch algorithm?
> >
> > I am also not sure about why does "untrained" matter. From my understanding, finding the winning ticket behaves similarly as training, and the pruning scores are just "trained" like the weights. Is "untrained" only of theoretical interest, or does it have any practical implications, such as faster convergence of the "training" algorithm?

---

> > > ### Author Response · Authors · 2020-11-23
> > > **Algorithmic details on the batchsize is in the appendix**
> > >
> > > Thanks a lot for giving us the opportunity to clarify your doubts.
> > >
> > > > In Alg. 1, the only loop is the epochs. There are no loop over minibatches. Is biprop a batch algorithm?
> > >
> > > In Algorithm 1, we have provided the pseudocode for “biprop” framework. In our implementation, as we use minibatch variant of the algorithm, there is another loop over minibatches. The implementation details are already highlighted in the appendix with details on the batchsize used. If reviewer suggests, we can add the minibatch loop in the Algorithm in the revised version. We skipped it in this version for clarity -- to reduce the number of notations in the pseudocode. We also plan to make our sourcecode open upon publication.
> > > Please let us know if this is not clear and we will be happy to clarify any other specific concerns you might have.
> > >
> > > >I am also not sure about why does "untrained" matter.
> > >
> > > As the term “untrained” is causing confusion, we will remove this term from the revised paper. Note that, our claims and contributions are not affected by using this term.
> > >
> > > >From my understanding, finding the winning ticket behaves similarly as training, and the pruning scores are just "trained" like the weights.
> > >
> > > As we have mentioned in Section 4: although we used gradient-based approaches in this paper to find MPTs, biprop is flexible to accommodate different class of algorithms that might avoid the pitfalls of gradient-based weight training.  Next, in contrast to weight-optimization that requires large model size and massive compute resources to achieve high performance, our hypothesis suggests that one can achieve similar performance without ever optimizing over weights of the large model.
> > > We would like to emphasize again that we have proposed new paradigm to learn binary neural nets and have made notable progress on three fronts as mentioned in the previous response.
> > >
> > > >Is "untrained" only of theoretical interest, or does it have any practical implications, such as faster convergence of the "training" algorithm?
> > >
> > > As mentioned above, the use of term “untrained” does not affect the claims and contributions of the paper. For a detailed discussion on the implication of the framework, please see Section 4 where we have also provided specific directions to be explored to achieve these implications.
> > > Also note that, both MPTs and FP weight-trained baselines, we used the same number of epochs as mentioned in the Appendix A.1. Theoretically, MPTs can be trained more efficiently than their FP counterparts. However, achieving this would require their implementation on specialized hardwares, e.g., FPGA, which is out of the scope of our paper.
> > >
> > > Please let us know if you have any further questions. We are hopeful that reviewer will see the contributions and potential of our approach.

---

> > > > ### Comment · AnonReviewer2 · 2020-11-24
> > > > **Still wondering with the advantage of finding lottery ticket over training**
> > > >
> > > > I am not sure if the authors can still response to the comments. I went through Sec. 4 again. The authors claimed " Next, in contrast to weight-optimization that requires large model size and massive compute resources to achieve high performance, our hypothesis suggests that one can achieve similar performance without ever training the large model."
> > > >
> > > > I understand that the resultant model can be compact and efficient for *inference*. However I am still not sure if it has any advantage for *training*. The biprop algorithm optimizes the pruning scores, which are still floating point numbers. From my understanding, the backprop of biprop still involves pure floating-point convolutions, and the binary weights do not accelerate the backprop. I am also not sure about the claim on "might avoid the pitfalls of gradient-based weight training". biprop itself is still a gradient-based algorithm. Does there exist any non-gradient algorithm for finding lottery ticket, that can achieve competitive performance with gradient-based algorithms?

---

> > > > > ### Author Response · Authors · 2020-11-24
> > > > > **Faster training approach is not the focus of the paper**
> > > > >
> > > > > We assume that R2 agrees with hypothesis-related and theoretical contributions of the paper that are independent of our algorithm related contribution, *biprop*, and he/she wants to understand advantages  provided by our algorithm. We note that *biprop* is one of many possibilities for identifying multi-prize tickets (MPTs) supported by the theory provided in our paper. Below is our response on the algorithmic contributions from two perspectives:
> > > > >
> > > > > 1. **What our implementation of *biprop* has already achieved**:
> > > > >     * Without significant hyperparameter tuning and without commonly used tricks in binary neural networks (BNNs), MPTs identified using *biprop* match or outperform BNNs trained using weight-optimization based methods. Further, MPT-1/32 are current state-of-the-art (SOTA) for binary-weight networks. We also note that additional hyperparameter tuning and regularization tricks from the BNN literature are expected to further improve these gains based on their effectiveness for existing BNNs. This already emphasizes the significant practical gain and potential of MPTs identified using *biprop*.
> > > > >     * Another aspect that was not the main focus of the paper and R2 seems to be interested in is to find out the training time comparison of *biprop* with weight-optimization of BNNs and full precision (FP) neural networks (NN). To compare *biprop* with weight-optimization of BNNs, we first note that gradient-descent based BNN weight-training usually requires significantly more epochs than FP NN. For example the BinaryConnect, ProxQuant, and DSQ baselines for CIFAR-10 (in Section 3.2) require 300-400 epochs. In comparison, our MPT-1/32 networks on CIFAR-10 were identified using *biprop* for 250 epochs. Therefore, informally we could say that in terms of train time: *biprop* < weight-trained BNN; in terms of accuracy we can say: MPT > BNN. So you could say our approach is faster to train than SOTA binary-weight methods as well as achieves higher accuracy. The reason we have not emphasized this in the paper is due to the fact that this was not the focus of the paper. Further, a carefully designed experimental study would need to be conducted to reliably report the comparison. For both MPTs and FP weight-trained baselines, we used the same number of epochs as mentioned in the Appendix A.1.
> > > > >
> > > > > 2. **What other “algorithmic implication or innovations” can be enabled to find MPTs**:
> > > > >     * Although we used a gradient-based approach in *biprop*, our general framework relying on pruning and quantization could potentially utilize gradient-free techniques. This would be significant to the binary neural network community as one of the pitfalls of gradient-based weight training is back-propagation through the sign acitvation function. However, this was not the focus of this paper but, instead, just one implication of our approach of pruning and quantizing to identify accurate BNNs.
> > > > >
> > > > > We would like to emphasize again that our main contribution (and focus) was not to design a faster training algorithm but instead to theoretically and empirically verify our multi-prize lottery ticket hypothesis. In this regard, we hope the reviewer can agree that *biprop* was successful. Any further implications or benefits afforded by *biprop* or our general framework of pruning and binarization as a training mechanism for BNNs provide promising future avenues of research for the deep learning and binary neural network communities.

---

> > > > > > ### Comment · AnonReviewer2 · 2020-11-24
> > > > > > **Overclaiming**
> > > > > >
> > > > > > I am not asking the paper to have a fast training algorithm. The reason I am asking is because the paper is claiming them, for example:
> > > > > >
> > > > > > "Next, in contrast to weight-optimization that *requires large model size and massive compute resources* to achieve high performance, our hypothesis suggests that one can achieve similar performance without ever training the large model"
> > > > > > - This is not well supported. The hypothesis only guarantees that there *exists* such solution. The solution is only achievable with a practical algorithm, which I assumes to be biprop. I think the claim can be only supported with a practical algorithm that achieve similar performance, which "doesn't require massive compute resources" or "doesn't need any training at all".

---

> > > > > > > ### Author Response · Authors · 2020-11-24
> > > > > > > **Reviewer is correct**
> > > > > > >
> > > > > > > We completely agree with the reviewer that we have not proved or showed "biprop" to be "not requiring massive compute resources".
> > > > > > >
> > > > > > > This was mentioned in the implication or future work section and we are sorry for not being precise when using this statement. We will make the change in the revised manuscript.
> > > > > > >
> > > > > > > We hope reviewer would agree that this was merely mentioned as an implication and not as the main contribution. Our contributions stand without this implication and the advantages of our approach w.r.t. existing approach, e.g., high accuracy, theoretical support, etc.

---

### Author Response · Authors · 2020-11-17
**Joint Response**

We thank all the reviewers for such a constructive review! We are encouraged that they found our idea to be novel (R1, R2), innovative (R4), practically significant (R1), as well as theoretically (R1, R3) and empirically (R2, R3, R4) strong. We were pleased to find that R1 and R3 realized the theoretical, empirical, and practical significance of our work and suggested accepting the paper (in R3's case, pending a clarifying response). While R2 and R4 recommended rejection, their main points of contention lie in the brevity of Section 2 (which we have now revised and extended by nearly two pages) which resulted in their interpretation that our work simply applies the original lottery ticket hypothesis to binary neural networks. In fact, we have not only proposed an entirely new hypothesis but our hypothesis (1) is stronger than the original lottery ticket hypothesis, (2) theoretically proved in our work, and (3) was empirically verified by conducting extensive experiments on small- and large-scale data sets (and achieving strong results). Additionally, we provide an efficient algorithm for testing our hypothesis. It is for these reasons we believe that our work is a more significant contribution than was initially precieved from the first draft of our paper. We hope that our revisions will clarify these points and that R2 and R4 will also champion our paper for acceptance.

We are very grateful for each reviewer's comments and perspective and have used their feedback to improve our paper. We have worked through our manuscript and have made several changes to the main body of the paper that provide more detail and insight into our theory and the algorithm. Additionally, we have strengthened our empirical results. In particular, we have made the following changes:

1. **Section 2.1 of the paper was revised and expanded to include the following**:
    a) Clearer statement of the theoretical result
    b) Proof sketch for the theoretical result
    c) Brief discussion and implications of theoretical result
2. **Section 2.2 of the paper was revised and expanded to include the following**:
    a) Thorough discussion and motivation for our method for identifying multi-prize tickets (MPTs)
    b) Detailed pseudocode for our algorithm used to empirically verify the Multi-Prize Lottery Ticket Hypothesis (biprop)
    c) Motivation for our choice of gradient estimator and new insight on its computational efficiency
3. **Updated empirical results in Section 3.2**:
    a) Improved accuracy and reduced parameter count of MPT-1/1 on CIFAR-10 by tuning hyperparameters in algorithm (biprop)
    b) Fixed a typo regarding BNN-1/1 SOTA on ImageNet (IR-Net): in the initial version we reported IR-Net to be achieveing 66.5% top-1 accuracy -- this was a typo and correct accuracy is 62.9% -- further reducing the gap between MPt-1/1 and SOTA.

Upon reflection, we believe that these changes have in fact made the paper stronger. We hope that these revisions and additions address all of your questions and resolve each of your concerns.


Please see answers to individual reviewer below, for particular comments.

--------

Based on a brief response by R2, we have made the following additional revisions so Section 2.2 to improve clarity: a) Added line numbers to Algorithm 1 and referenced these numbers in discussion of Algorithm 1, b) mentioned use of minibatches by Algorithm 1. Additionally, we have removed the use of the terminology "untrained" throughout the paper. However, we note that our claims and contributions are not affected by removing this terminology.

---

### Author Response · Authors · 2021-03-17
**Camera Ready Version**

We have now uploaded the camera ready version of our paper. A summary of the updates and additions for the camera ready version are provided below. Finally, we want to thank the reviewers and area chairs for their discussion during the review process and we are grateful for the acceptance of our work to ICLR 2021.

* **Updates**:
    * Updated results for Section 3.2 of paper including
        *  MPTs-1/32 not only set new binary weight network state-of-the-art (SOTA) Top-1 accuracy -- 94.8% on CIFAR-10 and 74.03% on ImageNet -- but also outperform their full-precision counterparts by 1.78% and 0.76%, respectively
        * MPT-1/1 achieves SOTA Top-1 accuracy (91.9%) for binary neural networks on CIFAR-10
    * Public release of biprop code and some pre-trained models are available at: https://github.com/chrundle/biprop
* **Additions**:
    * Enabled training of BatchNorm layer parameters while using biprop which boosts performance of MPT networks. Results for these networks are provided in Section 3.2

---

### Decision · Program_Chairs · 2021-01-07
**Final Decision**

**Decision:**

Accept (Poster)

**Comment:**

The authors present a new theoretical framework that establishes that any network can be approximated by pruning a polynomially larger random binary networks, and also an algorithm for pruning binary nets. The results are important in the general context of the "strong" lottery ticket hypothesis, and are of both theoretical and practical interest. Although some of the ideas and technical contributions can be seen as a combination of prior tools and algorithms, the experimental findings are very novel.  Some further concerns of clarity and novelty were addressed by the authors.